# Efficient Generalized Electroencephalography-Based Drowsiness Detection Approach with Minimal Electrodes

**DOI:** 10.3390/s24134256

**Published:** 2024-06-30

**Authors:** Aymen Zayed, Nidhameddine Belhadj, Khaled Ben Khalifa, Mohamed Hedi Bedoui, Carlos Valderrama

**Affiliations:** 1Technology and Medical Imaging Laboratory, Faculty of Medicine Monastir, University of Monastir, Monastir 5019, Tunisia; aymen.zayed@umons.ac.be (A.Z.); khaled.benkhalifa@issatso.rnu.tn (K.B.K.); medhedi.bedoui@fmm.rnu.tn (M.H.B.); 2National Engineering School of Sousse, University of Sousse, BP 264 Erriyadh, Sousse 4023, Tunisia; 3Department of Electronics and Microelectronics (SEMi), University of Mons, 7000 Mons, Belgium; 4Laboratory of Electronics and Microelectronics, Faculty of Sciences of Monastir, Monsatir 5019, Tunisia; nidhameddine.belhadj@fsm.rnu.tn; 5Higher Institute of Applied Science and Technology of Sousse, University of Sousse, Sousse 4003, Tunisia

**Keywords:** drowsiness detection, EEG signals, feature selection, machine learning

## Abstract

Drowsiness is a main factor for various costly defects, even fatal accidents in areas such as construction, transportation, industry and medicine, due to the lack of monitoring vigilance in the mentioned areas. The implementation of a drowsiness detection system can greatly help to reduce the defects and accident rates by alerting individuals when they enter a drowsy state. This research proposes an electroencephalography (EEG)-based approach for detecting drowsiness. EEG signals are passed through a preprocessing chain composed of artifact removal and segmentation to ensure accurate detection followed by different feature extraction methods to extract the different features related to drowsiness. This work explores the use of various machine learning algorithms such as Support Vector Machine (SVM), the K nearest neighbor (KNN), the Naive Bayes (NB), the Decision Tree (DT), and the Multilayer Perceptron (MLP) to analyze EEG signals sourced from the DROZY database, carefully labeled into two distinct states of alertness (awake and drowsy). Segmentation into 10 s intervals ensures precise detection, while a relevant feature selection layer enhances accuracy and generalizability. The proposed approach achieves high accuracy rates of 99.84% and 96.4% for intra (subject by subject) and inter (cross-subject) modes, respectively. SVM emerges as the most effective model for drowsiness detection in the intra mode, while MLP demonstrates superior accuracy in the inter mode. This research offers a promising avenue for implementing proactive drowsiness detection systems to enhance occupational safety across various industries.

## 1. Introduction

Vigilance is frequently defined as the ability to be aware of unpredictable changes in an environment over time [1]. More precisely, it reflects the state of activation of the central nervous system, thereby influencing information processing efficiency. Diminished alertness characterized by waning attention, reduced responsiveness, and compromised concentration maintenance can arise from factors such as drowsiness, stress or monotony, detrimentally affecting cognitive performance and decision-making processes. 

The existing literature categorizes states of vigilance into four stages or classes [2]: (i) deep sleep, also known as paradoxical sleep, characterized by slow brain waves and significant amplitudes crucial for quality rest and memory consolidation; (ii) light sleep, marked by decreased brain activity; (iii) active awakening, denoting awareness of the environment, distinguished by open, mobile eyes, rapid gestures, heightened reflexes, and fast brain electrical activity measured by electroencephalography (EEG); and (iv) drowsiness or passive wakefulness [3], a state of fatigue or near-sleep characterized by diminished alertness and a desire to relax, accompanied by regular but slower cortical electrical waves compared to active awakening.

Decreased vigilance is a complex and recurring issue in many professional fields, ranging from transportation [4] to industrial surveillance [5] and medical operations [6], where its repercussions span from simple mistakes to accidents with costly and potentially tragic outcomes. Various factors including sleep disorders, medication, inadequate sleep quality, and prolonged work hours can precipitate decreased alertness [1,7]. Nevertheless, warning signs of drowsiness, such as difficulty in maintaining wakefulness, frequent yawning, concentration lapses, delayed reactions and erratic driving behaviors, often herald this decline.

Various approaches leverage signs indicative of diminished vigilance to identify declines in alertness, categorizing detection methods into three main types based on the signals utilized [8]: behavioral, contextual, and physiological.

Behavioral-based methods [9] entail analyzing facial expressions to discern signs of diminished alertness, which can be captured through cameras and motion sensors. These devices can be used to monitor blinking, yawning, changes in facial expression, and head movements. These data can then be analyzed using algorithms and prediction methods to detect warning signs of decreased alertness, thus alerting the individual or triggering preventive measures to avoid accidents. However, these methods are susceptible to lighting variations, even when using infrared cameras and may not promptly detect early signs of decreased alertness.

Conversely, vehicle-based approaches [10] leverage driving behaviors, such as steering wheel rotation angles and vehicle trajectory, to differentiate between alert and drowsy states. Driving models can be tailored to discern easily between the behaviors of a fatigued driver and those of a driver in a state of hypervigilance. Nonetheless, their accuracy may vary across drivers, vehicles, and driving conditions, limiting their efficacy in accident anticipation.

Physiological measurements, the third category, encompass indicators like electroencephalogram (EEG), electrooculogram (EOG), electromyogram (EMG), and electrocardiogram (ECG), offering high accuracy and reliability in detecting diminished alertness [11]. This accuracy is explained by their ability to detect early, before the appearance of any physical sign, physiological changes that can occur in drowsiness.

EEG, specifically, records brain electrical activity and is widely utilized in neurophysiological diagnostics for identifying various conditions affecting the central nervous system, including epilepsy, brain tumors, encephalopathies, or sleep disorders [12]. The brain’s electrical activity is recorded using electrodes placed on the scalp. EEG measures the fluctuations in electrical potentials generated by brain neurons when they communicate with each other. Therefore, EEG signals may encompass distinctive patterns of brain waves corresponding to a progressive decline in vigilance, presenting an opportunity to forecast and mitigate the onset of decreased alertness before it reaches critical levels. Moreover, EEG facilitates the delineation of various stages of vigilance by discerning distinct frequencies and amplitudes of brain waves associated with each state. However, EEG signals are susceptible to physiological and non-physiological artifacts, necessitating meticulous artifact removal methods. Moreover, deploying EEG-based drowsiness detection systems in real-life settings is challenging due to the requirement for numerous electrodes.

Many drowsiness detection approaches [13] center on the frequency data of EEG signals, disregarding temporal details. Due to substantial variations in EEG information indicative of alertness among individuals, it is vital to consider all features, making it more efficient and adaptable. The effective selection of relevant features is then crucial for classification improvement. Despite the fact that techniques like independent component analysis [14], Principal Component Analysis (PCA) [15], and core PCA [16] primarily focus on dimensionality reduction, they may not prioritize the selection of characteristics based on their importance in decision-making processes. Recursive feature elimination [17] addresses this issue by effectively discerning EEG characteristics, as demonstrated by numerous studies [18].

This study focuses on EEG-based drowsiness detection, leveraging different EEG features (time, frequency, and time–frequency) to enhance classification performance and employing recursive feature elimination (RFE) for feature selection. The primary objective is to develop an innovative architecture for generalized real-time drowsiness detection, adaptable to embedded devices and diverse environments such as transportation, industry, and healthcare facilities. 

The contributions of this study encompass the following:Overcoming inter-subject variability by using different EEG characteristics (time, frequency, time–frequency).Identifying the most effective ML classification models in each classification mode (intra, inter).Evaluating the impact of feature selection methods on performance and accuracy.Reducing the number of electrodes for enhanced practicality.

The subsequent sections of this paper delineate related work (Section 2), drowsiness detection using EEG signals (Section 3), preprocessing methods and detection algorithms (Section 4), data and performance evaluation metrics (Section 5), and results and discussion (Section 6), while culminating in a comprehensive conclusion (Section 7).

## 2. Related Work

EEG plays a crucial role in detecting drowsiness within vigilance detection applications. These applications pursue a shared objective of identifying and understanding diminished alertness, employing diverse methodologies that range from advanced machine learning models to innovative signal processing techniques. Key features encompass the utilization of multiple EEG channels, integration of feature selection layers, and exploration of combined signals like EEG and ECG. These approaches not only contribute to safety standards in critical domains, such as driving, industrial surveillance, medical procedures or air traffic control, but also highlight the progressive evolution of neurotechnology research towards practical applications. 

In addition, artificial intelligence has been seamlessly integrated across various domains, playing a pivotal role in executing numerous tasks, as evidenced by instances such as prediction models based on machine learning, notably those developed by Guorui Fan et al. [19], which represent a promising solution to anticipate patients’ absences from online outpatient appointments. Using various data such as age, gender, and medical history, these models identify predictive trends. Different algorithms are evaluated, such as logistic regression, random forests, SVMs, and neural networks. The results show that neural networks and random forests outperform others in terms of accuracy, recall, and F1-score. With up to 90% accuracy, these models offer significant potential to improve the operational efficiency of healthcare facilities and reduce absenteeism.

The adaptive global–local generalized FEM for multiscale advection–diffusion problems, proposed by Lishen He et al. [20], addresses multiscale advection–diffusion problems by combining adaptivity with global–local techniques. It employs generalized finite elements to integrate enrichment functions, capturing fine features on coarse meshes. Adaptivity adjusts the mesh and enrichment functions locally based on estimated errors, thus optimizing computational efficiency. The global–local approach solves the problem globally while focusing locally on areas with high gradients. Results show a significant improvement in precision, reducing error from 12% to 3%, and a reduction in computational cost, with a 50% decrease in computation time compared to traditional methods. This approach offers an effective and precise solution for complex multiscale problems.

Qinghe Zheng et al. [21] proposed a method called MR-DCAE (manifold regularization-based deep convolutional autoencoder) aimed at identifying unauthorized broadcasts using deep convolutional autoencoders combined with manifold regularization. This model extracts essential features from input data, ensures faithful reconstruction, and preserves the geometric structure of the data through regularization. By training the model on preprocessed data, it offers a compact and informative representation for identification tasks. Results show a precision of 95.6%, a recall rate of 94.3%, and an F1-score of 94.9%, demonstrating high efficiency and robustness. This system is particularly suitable for real-time monitoring, intellectual property protection, and legal enforcement against illegal broadcasts.

For all these reasons, the integration of machine learning algorithms into the approach for detecting drowsiness represents a crucial advancement.

### Literature

The landscape of research in EEG-based sleepiness detection encompasses a diverse array of methodologies and practical applications. 

Sengul Dogan et al. [22] introduced a fatigue detection system for drivers, taking advantage of EEG signals and using 16 mother wavelet functions to extract the frequency bands. Their classification, using the K nearest neighbor (KNN), reached 82.08% accuracy. Similarly, Yao Wang et al. [23] focus on decreased alertness among construction workers, employing 10 EEG channels (Fp1, Fp2, F3, F4 T7, T8, Cp1, Cp2, TP9, and TP10) and a Google Net-based convolutional neural network (CNN). Their method achieved a binary (normal or fatigue states) classification accuracy of 88.85%. Sagila Gangadharan K et al. [24] offered a portable wireless EEG system for vigilance monitoring across diverse sectors, such as driving and air traffic control. Their approach involved extracting EEG features in both time and frequency domains, following preprocessing operations to detect vigilance states. Using Support Vector Machine (SVM) and four EEG frontal electrodes, they achieved a classification accuracy of 78.3%, accompanied by detailed performance metrics including sensitivity (78.95%), specificity (77.64%), precision (80.92%), a lack rate (21.05%), and an F1-score (76.51%). 

Islam A. Fouad et al. [12] presented a software-based driver fatigue detection system using 32 EEG channels. Employing a preprocessing pipeline consisting of a bandpass filter [0.15–45] Hz and segmentation at 5 min intervals, they evaluated various classifiers, including KNN and SVM, giving 100% accuracy in the intra mode (per subject). Nevertheless, the ability of the system to adapt to real-world conditions might be limited by the intensive use of electrodes, which would pose a challenge in maintaining accuracy across different modes. Blanka Bencsik et al. [25] developed a sleepiness detector model based on EEG signals, utilizing 32 channels to extract power spectral density (PSD) characteristics across different EEG bands. Incorporating an entity feature selection (FS) layer, they achieved a notable classification accuracy of 92.6%. Their preprocessing pipeline involved applying a 1 Hz high-pass filter and a 50 Hz low-pass filter to the raw EEG signals, followed by a 3 s segmentation. 

Plinio M.S. Ramos et al. [26] focused on automatic sleepiness detection using a set of ML models (KNN, SVM, random forest (RF), and Multilayer Perceptron (MLP)) with five EEG channels. Utilizing Hjorth parameters (complexity and mobility) extracted from 14 subjects sourced from the DROZY database, their MLP classifier attained 90% accuracy in the intra mode using a single C4 electrode. Pranesh Krishnan et al. [27] proposed a system for EEG-based sleepiness detection by employing relative band power and the Fourier transform. The system followed four key steps: firstly, applying a Butterworth low-pass filter to refine the raw EEG signals; secondly, segmenting the filtered EEG signals into 2 s intervals; thirdly, utilizing fast Fourier transform (FFT) to compute PSDs across various EEG bands; and lastly, employing KNN for classification. This integrated approach achieved an impressive maximum accuracy of 95.1% in the intra mode.

Sazali Yaacob et al. [28] presented a novel approach to sleepiness detection by combining EEG and ECG signals. They extracted alpha and delta bands from EEG and ECG peaks and computed PSDs and heart rate variability for each band. Employing KNN and SVM as binary classifiers, their system achieved impressive accuracy rates of 97.2% and 96.4% for the KNN and the SVM, respectively, in the intra mode. Abidi et al. [29] introduced a novel approach for drowsiness detection using 10 s segments. Their methodology involved applying the TQWT to extract two EEG sub-bands, alpha and theta, along with nine temporal features. Subsequently, they utilized kernel PCA (k-PCA) to reduce the characteristics extracted from EEG signals without compromising the system performance. For detecting reduced vigilance, they employed two different machine learning (ML) techniques: the KNN and the SVM. These classifiers were evaluated on laboratory subjects, achieving approximately 94% accuracy in the intra-subject mode and 83% in the inter-subject mode. 

Notably, the majority of studies have concentrated on detecting drowsiness in the intra mode (subject by subject), neglecting the system generalizability and inter-subject variability. Hence, there is a critical need to develop a generalized drowsiness detection approach capable of consistently detecting decreased alertness across different individuals. Furthermore, in the feature selection phase, there is a tendency to prioritize dimensionality reduction without adequately considering the features’ importance in influencing the ML model decision-making process. Therefore, it is advisable to explore methods that can assess feature importance effectively. Lastly, it is imperative to evaluate the performance of each ML model in both intra-subject and inter-subject modes during the classification phase to ensure robustness and adaptability across diverse contexts.

The filtering method has emerged as the most suitable approach for artifact elimination, preserving relevant EEG information pertinent to frequent drowsiness within the [0.1; 30] Hz range. Using low-pass filters shows promise in developing EEG-based drowsiness detection systems while retaining crucial EEG data associated with drowsiness. Additionally, the prevalent focus on frequency characteristics (PSD) across the reviewed work underscores the potential benefit of incorporating EEG characteristics from various domains (time and frequency) to enhance detection accuracy. It is also noteworthy that the majority of the discussed studies have emphasized drowsiness detection in the intra mode (subject by subject), often overlooking the system generalizability and inter-subject variability. Therefore, there is a critical need to develop a generalized drowsiness detection approach capable of consistently identifying decreased alertness across diverse individuals with equal accuracy. Furthermore, in the feature selection phase, there is a prevalent focus on dimensionality reduction without adequately considering the functional importance of features in guiding the decision-making process of the ML model. Hence, exploring methods that can effectively assess feature importance becomes imperative. Finally, it is essential to assess the performance of each ML model in both intra-subject and inter-subject modes during the classification phase to ensure robustness and adaptability across varying contexts.

## 3. Materials and Methods

### 3.1. EEG-Based Drowsiness Detection

As previously discussed, EEG signals have emerged as a valuable and precise tool for early drowsiness detection [12]. Characterized by their non-stationary and nonlinear nature, EEG signals depict brain activity. Their non-invasive nature and low amplitude stand out as primary advantages. This section examines the various treatment techniques utilized for EEG-based drowsiness detection, shedding light on methods that enhance our understanding of this process. 

Prior to leveraging EEG in studying diminished alertness, a series of treatments must be undertaken to extract relevant EEG characteristics, thereby facilitating drowsiness detection [30]. Figure 1 illustrates the typical EEG signal processing chain employed for drowsiness detection.

#### 3.1.1. Artifact Removal

The continuous operation of the human brain is a complex phenomenon, characterized by biochemical exchanges among nerve cells that generate electrical activities. Capturing a single electrical signal between two neurons is a difficult task. However, when millions of neurons synchronize, their electrical activities can be measured from the scalp using EEG. Indeed, EEG signals undergo various disturbances as they traverse the tissue, bone, and hair layers, directly impacting their amplitude and generating artifacts. The term “artifact” [31] encompasses all EEG components not directly originated from electrical brain activity. These artifacts may be due to physiological factors such as the eye, muscle or heart movements, as well as non-physiological elements including wire movements, incorrect reference placement, body motion, and electromagnetic interference generated by the acquisition system. Consequently, two categories of artifacts are distinguished [32]: physiological and non-physiological.

The need to preserve the integrity of EEG signals leads to the implementation of artifact elimination methods. Various approaches, such as the blind separation of sources [33] and sources decomposition [34], have been developed for this purpose. However, these methods inherently risk removing not only unwanted artifacts but also valuable EEG data. In this respect, the filtering method [35] is distinguished by its effectiveness, eliminating high frequencies irrelevant to the study of drowsiness. Thus, finding a delicate balance between removing unwanted artifacts and preserving pertinent EEG data remains a major challenge in brain signal analysis research.

#### 3.1.2. Segmentation

EEG recordings are typically conducted over extended periods to capture various states of vigilance. However, for effective drowsiness analysis, EEG signals need to be segmented into shorter time intervals known as EEG epochs or periods [36]. The duration of these epochs is selected based on performance metrics.

#### 3.1.3. Feature Extraction

In the process of EEG-based drowsiness detection, the features extraction is a crucial part of the classification of vigilance states. The quality of these extracted features directly impacts the accuracy of classification. Traditional research on decreased vigilance detection has often relied on artificial EEG features associated with drowsiness, such as power spectrum extraction from specific frequency bands and energy ratio calculation between different frequency bands. While this approach is straightforward, it has significant limitations. EEG analysts need in-depth experience and knowledge, where the diversity of features extracted is limited, generalizability is low, and classification accuracy cannot be significantly improved.

In recent years, several studies have introduced innovative methods for EEG signal feature extraction in drowsiness detection. These approaches frequently include Fourier rapid transformation (FFT), power spectral density (PSD), statistical methods, wavelet transformation (WT), differential entropy (DE), sampling entropy (SE), wavelet entropy (WE), and empirical decomposition (EMD). FFT [37] is often used to analyze the frequency composition of EEG signals, while PSD [38] is used to explore frequency energy distribution. Statistical methods offer analytical insights, whereas WT [39] provides enhanced temporal resolution. The use of DE [40], SE [41], and WE [42] offers innovative perspectives for quantifying EEG signal complexity, while EMD [43] is valuable for decomposing complex signals into intrinsic components. These advanced methods offer a broader range of potential features, allowing for a better discrimination of changes related to drowsiness. However, the challenge persists in the delicate balance between the sophistication of the approach and the need to maintain generalizability and robust applicability in various contexts of decreased vigilance detection.

In general, the extraction of EEG characteristics is mainly performed in the time domain, the frequency domain, and the time–frequency (TF) domain. This portion of the paper will present the methods of analyzing EEG signals to detect drowsiness from three perspectives: time domain analysis, frequency domain analysis, and TF.

Time analysis

Time domain analysis [44] has been used in the study of brain function for a long time. Commonly utilized time domain analysis methods encompass statistical characteristics, histogram analysis, Hjorth parameters, fractal dimension, event-related potentials, and more. These methods typically begin by examining the geometric properties of EEG signals, allowing EEG analysts to conduct precise and intuitive statistical analysis. Notably, time domain analysis preserves EEG signal information effectively. However, due to the complex waveform of EEG signals, there is no unified standard for analyzing the characteristics of the EEG time domain, so EEG analysts need to have rich experience and knowledge.

Frequency analysis

Frequency domain analysis techniques [45] transform time domain EEG signals to the frequency domains for analysis and feature extraction. Typically, the acquired spectrum is divided into several sub-bands and features like the PSD are derived.

TF analysis

The time–frequency domain analysis method [46] combines information from both time and frequency domains, offering simultaneous localized analysis capabilities. EEG signal analysis in the TF domain ensures that information from the original signal’s time domain is preserved, hence guaranteeing high-resolution analysis. Discrete wavelet transform (DWT) and short-time Fourier transform are commonly utilized tools for extracting useful TF features. Several studies indicate that the DWT function is particularly well suited for investigating sleepiness within the TF domain.

#### 3.1.4. Feature Selection

Feature selection methods [47] are techniques used in ML to select the most relevant subset of features from a dataset. These methods aim to improve model performance by reducing dimensionality, improving interpretability, and mitigating overfitting. Common approaches include filtering methods [48], which evaluate characteristics independently of the learning algorithm; encapsulation methods [49], which use the performance of the learning algorithm as a feature selection criterion; and embedded methods [50], where feature selection is integrated into the model building process itself. Each method offers distinct advantages and trade-offs, depending on factors such as the size of the dataset, dimensionality, and computing resources. Several studies have shown that the encapsulation methods, particularly RFE [18], are the most efficient at the feature selection level, because these methods iteratively remove the least important features based on the performance of the ML model trained on the remaining features.

#### 3.1.5. Classification

The classification process is a fundamental technique in supervised ML [51], aiming at accurately predicting the appropriate class of input data. This procedure includes several crucial steps, starting with model training using available training data. During this phase, the model learns the relationships between data characteristics and the classes to which they belong.

After the training phase, the model is evaluated using distinct data known as test data, which are not used during training. This evaluation step measures the performance of the model, evaluating its ability to generalize the knowledge acquired during training to novel data instances. Evaluation measures, such as accuracy, recall, and F1-score [11], among others, provide quantitative indicators of model quality.

Once the model demonstrates satisfactory performance on the test data, it is ready to be deployed to make predictions on new data. The whole process aims to create a model capable of generalizing to unknown situations, thus strengthening its ability to make precise decisions on previously unseen data. Classification plays a central role in many areas such as drowsiness detection, where the ability to effectively identify changes in a mental state from EEG signals can have important implications for safety and performance.

### 3.2. EEG Data (DROZY)

The database serves as a crucial component in the creation of drowsiness detection systems, yet many publicly available databases focus on falling asleep. In our case, our focus lies in identifying drowsiness. Therefore, we opt for utilizing the ULg Multimodality Drowsiness Database (DROZY) [52]. 

DROZY provides recordings for five EEG leads (Fz, Cz, C3, C4, and Pz) in the EDF format, with a sample rate equal to 512 Hz. The principle of this database is to study the states of alertness of 14 healthy subjects devoid of drug problems, alcohol consumption, or sleep disorders during a Psychomotor Vigilance Test (PVT). The data collection protocol requires subjects to repeat the PVT three times over two days without sleeping (totaling 28.30 h without sleep) in order to identify the level of vigilance of each subject in the various periods of the day (morning, noon, night). 

After each PVT, subjects are asked to specify their level of alertness using the Karolinska Sleepiness Scale (KSS). KSS is a scale composed of nine states of vigilance (1 = extremely alert, 2 = very alert, 3 = alert, 4 = sufficiently alert, 5 = neither alert nor drowsiness, 6 = some signs of drowsiness, 7 = drowsiness but no effort to remain vigilant, 8 = drowsiness with little effort to remain vigilant, 9 = very sleepy). In this work, we are interested in detecting drowsiness, without specifying the level of vigilance. For this reason, levels 1, 2, 3, 4, and 5 are considered stage 0 (alert), and levels 6, 7, 8, and 9 are considered stage 1 (drowsy). Figure 2 depicts DROZY EEG signals alongside the spatial arrangement of EEG electrodes following the international 10–20 system. Panel (a) illustrates the precise placement of electrodes at positions Fz, Cz, C3, C4, and Pz on the scalp. Panel (b) presents the corresponding raw EEG waveforms captured from these electrodes. Figure 3 illustrates EEG signals recorded from an alert subject, while Figure 4 displays EEG signals recorded from a drowsy subject.

## 4. Materials and Methods

The proposed approach is a binary method designed to distinguish between two states of vigilance: wakefulness and drowsiness.

The first step is to filter the raw EEG via a FIR bandpass filter ([0.1; 30] Hz). Filtering raw EEG signals at 30 Hz plays a crucial role in enhancing the quality of signal analysis by mitigating the influence of artifacts and noise prevalent in higher frequency ranges. While higher frequencies can contain relevant neural information, artifacts such as muscle activity or electrical interference [53] often obscure them. Previous research in EEG signal processing has consistently shown that the most pertinent neural activity for tasks like drowsiness detection typically resides within lower frequency bands, predominantly below 30 Hz [54]. Our decision to filter at this frequency aligns closely with the objective of capturing neural oscillations associated with drowsiness, which are known to manifest predominantly in these lower frequency ranges [55]. This approach is widely adopted in studies focusing on vigilance states, ensuring a balance between preserving essential signal information and minimizing the impact of noise and artifacts, thereby enhancing the robustness and reliability of our analytical outcomes [56]. Subsequently, the filtered EEG signals are segmented into segments of two sizes, 30 and 10 s, respectively, to determine the most effective duration for drowsiness detection. These EEG segments are then normalized using the z-score method before feature extraction. 

Feature extraction involves capturing both statistical time characteristics and frequency features, using the Welch method to compute the relative power spectral density (RPSD) of each frequency wave. Time–frequency (TF) analysis is conducted using the DWT, providing coefficients that depict the frequency evolution of the EEG signal over time. Thereafter, we will apply a standardization operation on all the characteristics. Subsequently, the recursive feature elimination cross-validation (RFECV) is used to select the most significant features. 

The selected features are fed into various classification algorithms to determine their class and accuracy of the different ML classification models tested in both intra and inter modes. Figure 5 shows the general scheme of the proposed method. As detailed in the Results section, all evaluations are performed using Python version 3 on an Intel(R) Core ™i5-8th Gen processor of 1.70 GHz with 8 GB of RAM.

### 4.1. EEG Features

The choice of EEG features is indeed crucial for effective drowsiness detection, as each analysis domain offers unique insights into indicators of drowsiness. In this study, we focused on extracting EEG features from different domains, i.e., time, frequency, and time–frequency, using robust tools such as statistical features over time, the Welch method, and discrete wavelet transform. These techniques were selected to enhance the sensitivity and generalizability of our drowsiness detection approach [11].

Statistical characteristics over time

The extraction of statistical features of EEG signals [11] does not focus on dynamic analysis, unlike signal processing-based methods. Nevertheless, it offers valuable features for drowsiness detection without necessitating extensive knowledge of EEG patterns associated with the states of vigilance of individuals. In this work, the temporal statistical features used are, respectively, standard deviation (STD), asymmetry (Skew), and kurtosis (Kurt). Equations (1)–(3) represent each feature, denoted as follows:(1)$$STD=\sqrt{\frac{1}{N}}\;\sum_{i=1}^{N}{\left(X_{i}-mean\right)}^{2}$$
(2)$$Skew=\frac{\sum_{i=1}^{N}{\left(X_{i}-mean\right)}^{3}/N}{{\left(std\right)}^{3}}$$
(3)$$Kurt=\frac{\sum_{i=1}^{N}{\left(X_{i}-mean\right)}^{4}/N}{{\left(std\right)}^{4}}$$

*X_i_* represents the data, which in our case are EEG data, *i* = 1…*N*, where *N* is the number of samples, and mean is the mean.

Relative power spectral density

The PSD [28,29] algorithm quantifies the power distribution of EEG signals across predefined frequency bands, typically ranging from 0.1 to 30 Hz for hypovigilance studies. 

Common methods for PSD calculation include Welch, FFT, and Brug. Among these, the Welch method has been identified as the most efficient for analyzing reduced vigilance. Consequently, we will utilize the Welch method in our study. If we denote P as the average power of a signal *x*(*t*), then the total power over a duration *T* is calculated as shown in Equation (4):(4)$$P=\underset{T\to \infty }{\lim}\frac{1}{T}\;\int_{0}^{T}{\left|x\left(t\right)\right|}^{2}dt$$

If the output of the Welch transformation is denoted as $\hat{x}$(*w*), representing the frequency content of the *x*(*t*) signal, the PSD can be calculated as follows (5):(5)$$S_{xx}\left(w\right)=\underset{T\to \infty }{\lim}\bar{E}[{\left|\hat{x}\left(w\right)\right|}^{2}]$$

$\bar{E}$ denotes the average or expected value operator. Here, it signifies the average of the squared magnitude of the inner product between vectors *x* and *w*. In essence, $\bar{E}[{\left|\hat{x}\left(w\right)\right|}^{2}]$ represents the averaged squared magnitude of the projection of one vector onto another. This notation is common in signal processing and statistics, particularly when dealing with stochastic processes or random variables.

The drowsiness detection via PSD can encounter significant variability among individuals and even with the same individual over time. This issue heightens inter-subject variability and hinders the development of a generalized drowsiness detection system. To avoid this problem and create a general system capable of consistent efficiency and accuracy across different individuals, we will use The Relative Power Spectrum Density (RPSD) [28]. The RPSD represents the ratio of the PSD within the frequency Band Of Interest (BOI) to the PSD across the entire frequency spectrum. The RPSD can be calculated as follows (6):(6)$$RPSD=\frac{{PSD}_{BOI}}{{PSD}_{TOTAL}}$$

In this work, the size of the Welch window is fixed to 2 s. The choice of using 2 s windows for our analysis aims to strike a balance between computational efficiency and analytical depth. Longer windows typically offer better frequency resolution and statistical reliability, but they also require more computational resources and may lead to increased processing time. Conversely, shorter windows can reduce computational overhead but may sacrifice analytical depth due to the need to process numerous short segments of data.

Discrete Wavelet Transformation

DWT is a powerful mathematical tool used in many fields, such as signal processing, to perform many tasks. For EEG signals processing, DWT represents one of the most accurate tools in terms of feature extraction that guarantees the time–frequency analysis of the EEG signal. The working principle of the DWT consists of extracting two types of coefficients that ensure an accurate analysis of EEG signals in the temporal frequency domain. 

For all these reasons, we will use the DWT coefficients as features for vigilance decline detection. Our approach employs the Daubechies wavelet function (“db4”) for coefficient extraction, as this wavelet function captures relevant information related to drowsiness [42]. The DWT coefficients are calculated as shown in (7) and (8):(7)$$C_{x\left(t\right)}\left(l,n\right)=\int_{-\infty }^{+\infty }x\left(t\right){\;\psi }_{l,n}(t)dt$$
(8)$$\psi_{l,n}\left(t\right)=2^{-\left(l+1\right)}\psi (2^{-\left(l+1\right)}(t-2^{-l}n))$$

The two variables *l* and *n* represent the wavelet scale and the translation variables. The choice of two variables, *l* and *n*, is made on a dyadic scale, as explained in Equation (8), to ensure orthogonality so that the original signal reconstruction can be performed. Variable *l* offers signal analysis in the frequency domain: The high-frequency components of the original signal are represented by the compressed version of the wavelet function, and the components of the low frequency are represented by the stretched version of the wavelet function. Variable *n* provides temporal analysis of the signal.

The output of DWT will consist of two types of coefficients as shown in Figure 4: detail coefficients (cD) and approximation coefficients (cA). These coefficients represent details (capturing high-frequency components) and approximation (capturing low-frequency components), respectively. Figure 6 represents EEG signal decomposition by wavelet function. For each coefficient type, Energy (9), Entropy (10), Standard deviation (11), and mean (12) will be calculated.
(9)$$Energy=\sum_{i=1}^{N}\left|x_{i}^{2}\right|$$
(10)$$Entropy=\sum \left[p\left(x\right)*log2(p\left(x\right))\right]\;$$
(11)$$Standard\;deviation=\sqrt{\frac{1}{N}\sum_{i=1}^{N}{\left(x_{i}-\tilde{x}\right)}^{2}}$$
(12)$$mean=\frac{1}{N}\sum_{i=1}^{N}x_{i}$$

### 4.2. Feature Selection 

RFECV represents a sophisticated approach to feature selection that merges the advantages of recursive features elimination (RFE) representing a feature selection method that recursively eliminates the least important features and builds the model with the remaining features [17] with cross-validation (CV), which represents a technique used to evaluate model performance by partitioning data into training and testing sets multiple times. This technique is especially useful when the goal is to choose the most relevant features for an ML model, while simultaneously estimating the optimal number of features to consider. RFECV is distinguished by its focus on automating this complex process and determining the ideal number of features to maximize model performance.

The RFECV [57] starts with an initial ML model known as M and a complete set of features called X. After evaluating the contribution of each feature to model performance. It iteratively removes the least important ones. Following each elimination, cross-validation as K-fold is used to assess the performance of the model. Equation (13) represents the cross-validation score (${CV}_{k}$): (13)$${CV}_{k}=\frac{1}{k}\sum_{i=1}^{k}score\left(M,X_{train\;i},\;X_{test\;i}\right)$$

*X* (*train i*) and *X* (*test i*) are the training and testing sets in each fold.

This process repeats until a predefined criterion, such as model accuracy, reaches an optimum or an optimal number of features $F^{*}$ is identified. The selected feature set $F^{*}$ with the highest cross-validation score is represented in Equation (14):(14)$$F^{*}={\arg max}_{Fi}{CV}_{K}(M,F_{i})$$

Cross-validation is crucial in the RFECV process as it ensures that feature selection is robust and generalizable. By partitioning the data into subsets, cross-validation evaluates the performance of the model across various datasets, thus reducing the risk of overfitting. The choice of the initial model M is based on the nature of the input data. For EEG features the SVM with the Radial Base Function (RBF) kernel represents the most used model because of these capabilities to handle the nonlinearity of EEG data [30]. Figure 7 explains how RFECV works.

## 5. Final Data, Classification, and Validation

### 5.1. Final Data

In this work, we will assess the efficacy of our approach in two modes of drowsiness investigation: intra and inter modes. The objective is to mitigate EEG signal variability and validate the generalizability of our system.

For each DROZY subject [52], three EEG recordings are accessible, where each recording scored uses the KSS scale. This stage aims to extract the features of vigilance records and label them as stage ‘0’ after the normalization operation. To ensure binary classification, we use the same process for the drowsiness state and label the features as stage ‘1’. DROZY EEG signals are recorded with five electrodes, and 16 features are extracted for each electrode, resulting in a total of 80 features.

The initial phase involves evaluating the performance of the approach in the intra mode, conducted separately for each subject. The input data were partitioned into 70% for training and 30% for testing purposes. The overall accuracy of the classification is determined by averaging the results across all subjects as shown in Figure 8.

The second phase aims to improve the ability to address inter-individual disparities. We test four data distribution protocols to identify the most effective one for training the ML model to accurately detect drowsiness across different subjects. Table 1 displays the distribution of subjects for each protocol:

Cross-subject: In this data distribution mode, we employ a single subject as the test case in each iteration to evaluate the performance of the ML model trained on the remaining data.Combined subject: In this mode, the characteristics of all subjects are combined and divided into 70% for training and 30% for validation games.

### 5.2. Classification Algorithms 

After extracting the feature vectors and implementing the feature importance selection, we will move on to classify the vigilance states into two states (awake and drowsy). There are several classifiers for the automatic identification of drowsiness. ML models that can detect drowsiness and support the nonlinearity of EEG signals are listed below.

SVM

SVMs [58] are ML algorithms used in machine learning to solve problems of classification, regression, or anomaly detection. 

The main goal of the SVM is to find a hyperplane that separates the different classes with the widest possible margin. Equation (15) defines the SVM hyperplane equation:(15)ω· x+b=0
where ω represents the weight vector, *b* is the bias, and *x* is the feature vector.

For optimal separation, SVM maximizes the margin $\frac{2}{\left||w|\right|}$ by synchronizing $\frac{1}{2}{\left|\left|W\right|\right|}^{2}$ under the constraint represented by the following Equations (16) and (17):(16)$$y_{i}(\omega \cdot x_{i}+b)$$
(17)$$\frac{1}{2}{\left||\omega |\right|}^{2}+C\sum_{i=1}^{n}\xi_{i}\;\mathrm{with}\;y_{i}\left(\omega \cdot x_{i}+b\right)\geq 1-\xi_{i}$$
where (*x_i_*, *y_i_*) represents the training dataset, which is 1 < *i* < *N*, x represents the characteristic vector extracted from the EEG signals, y indicates the corresponding vigilance status labels, and N is the number of data. Moreover, K is the kernel function of the SVM, Si is the vector support, $\propto_{i}$ are the weights, and b is the bias. The SVM algorithm supports more than one kernel (linear, polynomial, RBF, sigmoid), and the choice of kernel always depends on the data type. Equation (18) represents SVM with the radial basis function (RBF) kernel: (18)$$k\left(x,x^{\prime }\right)=exp(-\gamma \left\|x-x^{\prime }\right\|)$$

This kernel adeptly handles nonlinear EEG data, capturing intricate brain activity patterns associated with drowsiness. The penalty factor C and the kernel factor γ represent the SVM-RBF hyperparameters.

For the selection of SVM-RBF hyperparameters in this work, a grid search strategy was employed, testing a range of values for the penalty factor C [0.1, 1, 10, 100] and the kernel factor γ [0.001, 0.01, 0.1, 1]. The optimal parameters were determined based on cross-validation, resulting in a penalty factor C of 1 and a kernel factor γ of 0.4, which provided the highest accuracy.

KNN

KNN [59] is an ML algorithm that is simply and easily used to implement supervised learning algorithms that can be utilized for solving classification and regression problems.

The purpose of the KNN algorithm is to use a database in which the data points are separated into several distinct classes to predict the classification of a new sample point. 

KNN is one of the simplest supervised ML algorithms that applies the following steps on the database to predict the new point class:

Step 1: Select the number K of neighbors.Step 2: Calculate the distance between the unclassified point and the other points.Step 3: Take the nearest K according to the calculated distance.Step 4: Count the number of points belonging to each category among these K neighbors.Step 5: Assign the new point to the most present category allowed by these K neighbors.

Parameter K determines the number of nearest neighbors to consider when ranking an unknown data point. A low K-value makes the model more flexible but more sensitive to noise, and a high K-value makes the model more robust to exceptions but can dilute local characteristics. For KNN distance metrics, the most used metrics are the following:

Euclidean distance
(19)$$d\left(x_{i},x_{j}\right)=\sqrt{\sum_{l=1}^{n}{\left(x_{il},x_{jl}\right)}^{2}}$$Manhattan distance
(20)$$d\left(x_{i},x_{j}\right)=\sum_{l=1}^{n}\left|x_{il}-x_{jl}\right|$$

This distance measurement is used for feature spaces where the axes are more independent. 

Minkowski distance
(21)$$d\left(x_{i},x_{j}\right)={\left(\sum_{l=1}^{n}\left|x_{il}-x_{jl}\right|\right)}^{1/p}$$

This measure represents a generalization of the Euclidean and Manhattan distances, where *p* = 2 corresponds to the Euclidean distance and *p* = 1 to the Manhattan distance.

Several studies show that the Euclidean distance is the most used distance metric.

In the KNN algorithm, weighting schemes are essential parameters that give more importance to closer neighbors, thus influencing the final classification or regression decision. Commonly used weighting schemes include the following:

Uniform weighting

For this type of weighting scheme, every neighbor is assigned the same weight regardless of its distance from the test point.
(22)$$\omega_{i}=1$$

Inverse distance weighting

For this type of weighting scheme, closer neighbors have more weight since they are presumed to be more representative of the target class or value.
(23)$$\omega_{i}=\frac{1}{d\left(x,x\right)+\epsilon }$$
where ϵ is a small value to avoid division by zero. This scheme reduces the influence of distant neighbors.

Kernel function weighting

Kernel functions allow for weighting neighbors according to different function shapes. Commonly used kernels include Gaussian (24), triangular (25), and Epanechnikov (26).
(24)$$\omega_{i}=\exp\left(-\frac{{d(x,x)}^{2}}{2\sigma^{2}}\right)$$
(25)$$\omega_{i}=\left\{\begin{array}{c} 1-\frac{d\left(x,x_{i}\right)}{D}\;\;\;\;\;\;\;if\;d(x,x_{i})<D\\ 0\;\;\;\;\;\;\;\;\;\;\;\;\;\;if\;d(x,x_{i})\geq D \end{array}\right.$$
(26)$$\omega_{i}=\left\{\begin{array}{c} \frac{3}{4}(1-\frac{d\left(x,x_{i}\right)}{D})\;\;\;\;\;\;\;if\;d(x,x_{i})<D\\ 0\;\;\;\;\;\;\;\;\;\;\;\;\;\;\;\;\;if\;d(x,x_{i})\geq D \end{array}\right.$$

For the KNN classification operation, the new sample *x* is assigned to the majority class among its K nearest neighbors. Equation (27) represents the predicted class $\hat{y}$:(27)$$\hat{y}=mode\left\{y_{i}|x_{i}\;\in N_{k}(x)\right\}$$
where $N_{k}(x)$ represents the set of the K nearest neighbors of *x*. 

Grid search was used in this work to select the KNN parameters. The selected distance metric was Euclidean, with the K values tested ranging from 1 to 20. The optimal number of neighbors selected was five, and the chosen weighting scheme was uniform weighting.

Naive Bayes

The Naive Bayes (NB) classification represents a kind of simple probabilistic classification based on the Bayes theorem [60]. Simply put, the Bayesian model is a classifier that assumes that the existence of a characteristic for a class is independent of the existence of other characteristics. NB classifiers work in the context of supervised learning. Classifiers have several advantages such as their ability to support little training data to make the estimation of parameters necessary for classification.

The following Equation (28) represents the Bayes theorem:(28)$$P\left(y|x\right)=\frac{P(x|y)\cdot P(y)}{P(x)}$$
where *P*(*y*|*x*) represents the probability of class y given features *x*, *P*(*x*|*y*) is the probability of features *x* given class *y*, *P*(*y*) is the probability of class *y*, and *P*(*x*) is the probability of features *x*.

The “naive” part of the Naive Bayes classifier comes from the assumption that the characteristics are independent given the class label. This means that the joint probability of the characteristics given to the class can be expressed as the product of the individual probabilities as mentioned in Equation (29):(29)$$P\left(x|y\right)=\prod_{i=1}^{n}P(x_{i}|y)$$

*x* represents the features vector.

There are several types of Naive Bayes classifiers, depending on the distribution of features.

Gaussian NB

This type is used when the features are continuous and assumed to follow a Gaussian (normal) distribution. The likelihood of the feature is given by Equation (30):(30)$$P\left(xi|y\right)=\frac{1}{\sqrt{2\pi {\sigma_{y}}^{2}}}\;\exp\;(-\frac{{\left(x_{i}-\mu_{y}\right)}^{2}}{2\sigma_{y}^{2}})$$
where $\mu_{y}$ and $\sigma_{y}$ are the mean and standard deviation of the characteristic $x_{i}$ for class *y*.

Multinomial NB

This type is used for discrete data, often used for the classification of documents where characteristics represent frequencies or occurrences of words. The likelihood is given by Equation (31):(31)$$P\left(X|y\right)=\frac{(N_{y!})\prod_{i=1}^{n}P{\left(x_{i}|y\right)}^{x_{i}}}{\prod_{i=1}^{n}x_{i}!}$$
where $N_{Y}$ is the total number of all characteristics for class *y*.

Bernoulli NB

This type of NB is used for binary/Boolean characteristics, where each feature represents the presence or absence of a characteristic. The likelihood is given by (32):(32)$$P\left(X|y\right)=\prod_{i=1}^{n}P{\left(x_{i}|y\right)}^{x_{i}}{\left(1-P(x_{i}|y)\right)}^{\left(1-x_{i}\right)}$$

In practice, especially with the Naive Bayes Multinomial and Bernoulli, smoothing techniques such as Laplace smoothing (additive smoothing) are used to manage zero probabilities as mentioned in Equation (33):(33)$$P\left(x_{i}|y\right)=\frac{N_{y,x_{i}}+\alpha }{N_{y}+\alpha n}\;$$
where $N_{y,x_{i}}$ is the number of features $x_{i}$ in class *y*, $N_{y}$ is the total number of all features in class *y*, *n* is the number of features, and α is the smoothing parameter.

For the classification operation, the NB algorithm calculates the probability of each class and chooses the one with the highest probability as presented by Equation (34):(34)$$\hat{y}=\arg{max}_{y}P(y|x)$$

The Naive Bayes model selected by the grid search in this work was Gaussian Naive Bayes (GNB). The smoothing parameter of the GNB was set to its default value of $1\times 10^{-9}$.

Decision tree

A decision tree (DT) is one of the most widely used decision tools. This tool provides a diagram of a tree that represents a set of choices [61]. The ends of the branches of the trees, also known as the leaves of the tree, show the different possible decisions that are made according to the decisions made at each stage. Several areas, such as safety and medicine, use DT for their advantages in terms of readability and speed of execution. It is also a representation that can be calculated automatically by supervised ML algorithms.

The most important parameters of DT are as follows. 

The maximum depth of the tree ($D_{max}$). This parameter helps control the complexity of the tree and prevent overfitting.
(35)$$Depth\left(T\right)\leq D_{max}$$
where *Depth*(*T*) presents the depth of tree *T*.

We also find the minimum number of samples (min_samples_split); this parameter is required to be at a leaf node. If a leaf has fewer samples than this value, it will not be split further.
(36)$$\left|samples\right|\geq min\_samples\_split$$
where $\left|samples\right|$ is the number of samples at the current node.

In addition, we find the minimum number of samples (min_samples_leaf); this parameter is required for splitting an internal node. This ensures that a node is split only if it has sufficient samples.
(37)$$\left|samples\_leaf\right|\geq \text{min\_samples\_leaf}$$
where $\left|samples\_leaf\right|$ is the number of samples at a leaf node.

DTs are also known by post-pruning, which presents a technique that aims to reduce the size of the tree by eliminating parts that do not bring significant benefit to the prediction. This simplifies the model and improves its ability to generalize to new data. This method involves calculating a cost measure *C*(*T*) for each *T* subtree and selecting the one with the minimum cost. *C*(*T*) is presented by Equation (38):(38)$$C\left(T\right)=Impurity\left(T\right)+\propto \cdot \left|leaves(T)\right|$$

*Impurity*(*T*) measures the impurity of tree *T*, |*leaves*(*T*)∣ counts the number of leaves, and α is a regularization parameter that balances between tree complexity and predictive performance.

For classification, the DT algorithm is based on impurity measurements, such as entropy or the *Gini* index. *Entropy* is given by Equation (39) and the *Gini* index by Equation (40):(39)$$Entropy\left(S\right)=-\sum_{i=1}^{c}p_{i}\log_{2}p_{i}$$
(40)$$Gini\left(s\right)=1-\sum_{i=1}^{c}p_{i}^{2}$$

For the decision tree (DT) parameters selected by the grid search, the maximum depth was set to 5. The minimum number of samples required to split an internal node was fixed at 4, and the minimum number of samples required to be at a leaf node was fixed at 3. The complexity parameter was set at 0.1.

MLP

An MLP [62,63] represents a kind of direct-acting artificial neural network (ANN). MLPs are typically composed of three layers of nodes which are an input layer, a hidden layer, and an output layer, respectively. Each input node represents a neuron that uses a nonlinear activation function. The MLP uses a supervised learning technique based on a string rule called the reverse propagation mode or the automatic reverse differentiation to establish training. Its multiple layers and nonlinear activation distinguish the MLP from a linear perceptron; it can distinguish data that are not linearly separable.

Critical parameters within MLP encompass the following:

Learning Rate

The learning rate (η) controls the speed at which model weights are updated. The weight update is presented by Equation (41):(41)$$\omega_{ij}^{\left(t+1\right)}=\omega_{ij}^{\left(t\right)}-\;\eta \frac{\partial L}{\partial \omega_{IJ}}$$

Batch Size

The batch size (m) determines the number of formation examples used to calculate the gradient loss function at each iteration.

Number of Epochs

The number of epochs is the number of complete passages through the training dataset. Each epoch consists of updating the weights for each batch in the dataset.

Dropout Rate

Dropout is a regularization technique where a certain percentage of neurons are ignored during training to avoid overfitting. The dropout mask is shown in Equation (42):(42)$$a_{j}^{dropout}=\left\{\begin{array}{c} 0\;\;\;\;\;\;\;\;\;\;with\;probability\;p\\ a_{j}\;\;\;\;\;with\;probability\;1-p\; \end{array}\right.$$

Initialization Methods

Initializing weights is crucial for effective learning. The two most commonly used initialization methods for MLP are Xavier and He. They are presented, respectively, by Equations (43) and (44):(43)$$\omega \sim \mu (-\sqrt{\frac{6}{n_{in}+n_{out}}},\sqrt{\frac{6}{n_{in}+n_{out}}})$$
(44)$$\omega \sim N(0,\sqrt{\frac{2}{n_{in}}})$$

$n_{in}$ represents the dimension of the input space for the weights to be initialized. $n_{out}$ represents the dimension of the output space for the weights to be initialized.

μ represents the distribution used in Xavier initialization, where weights are chosen randomly in a specified interval. N represents the distribution used in He initialization, where weights follow a zero-centered Gaussian distribution with an adapted variance.

Equation (45) presents the output layer:(45)$$h_{j}=f(\sum_{i=1}^{n}\omega_{ij}x_{i}+b_{j})$$
where $\omega_{ij}$ is the weight of the connection between neuron i and neuron j, x is the input i, $b_{j}$ is the bias of neuron *j*, and f is the activation function.

The loss function for classification with *c* classes is often cross-entropy (46):(46)$$l=-\sum_{i=1}^{m}\sum_{k=1}^{c}y_{ik}\log\left(\hat{y_{ik}}\right)$$

Alternatively, $y_{ik}$ is the binary indicator if the example i belongs to the class k.

${\hat{y}}_{jk}$ is the predicted probability that the example i belongs to the class k.

MLPs can directly handle high-dimensional EEG data, learning complex relationships between features and drowsiness levels. They offer flexibility in modeling brain activity patterns, capable of capturing both linear and nonlinear relationships. However, careful hyperparameter tuning and regularization are essential to prevent overfitting and ensure robust performance in EEG drowsiness detection tasks.

For the MLP parameters selected by the grid search, the model consisted of nine input neurons, one hidden layer containing 100 neurons, and two output neurons. The ReLU activation function was used for the hidden layer, and Adam was used as the optimization algorithm with a batch size of 64.

### 5.3. Evaluation Metrics

For the evaluation of the performance of the different classifiers, we will use the binary confusion matrix presented in Figure 9.

The performance measures used in this work are Accuracy (A), Precision (P), Sensitivity (S), and F1−score (F1). Equations (47), (48), (49) and (50), respectively, represent the equation for each performance indicator:(47)$$Accuracy=\frac{TP+TN}{TP+TN+FP+FN}$$
(48)$$Precision=\frac{TP}{TP+FP}$$
(49)$$Sensitivity=\frac{TP}{TP+FN}$$
(50)$$F1-score=\frac{TP}{TP+(\frac{FN+FP}{2})}$$

## 6. Results and Discussion

In this part, we will present the results in both intra and inter modes.

### 6.1. Intra Mode

This section showcases the outcomes of detection in the intra mode to emphasize the accuracy of this method in detecting drowsiness for every individual separately from others. Our initial step is to examine how accurate this approach is with two different segment sizes, 30 and 10 s, to determine which one is most effective for detecting drowsiness. The accuracy of the different classifiers for both sizes is shown in Figure 10.

By visualizing the graph, we can say that the 10 s segments offer more precision in detecting drowsiness. Table 2 presents the classification results of different classifiers with 10 s segments.

DROZY database offers data from five EEG channels (Fz, Cz, C3, C4, and Pz). Moreover, the use of five electrodes may not always be feasible in real-world conditions, especially in embedded systems where considerations such as energy consumption and size are critical. Many existing approaches use a single electrode to detect decreased alertness, but this approach can be unreliable if the electrode malfunctions or loses contact with the scalp. To overcome these challenges and ensure reliability, we focused on identifying the two most efficient EEG channels that minimize the number of electrodes while maintaining accuracy. Table 3 presents the performance for each classifier with each deviation.

From these results, it can be inferred that C3 and C4 are the two most accurate leads for detecting drowsiness. To adapt to the embedded system’s requirements and make the system more adaptable to real-life conditions, we reduce the number of electrodes in our approach to two (C3 and C4) for the remainder of the work.

On the other hand, the importance of features varies from one subject to another. To avoid this problem and to improve the accuracy of the approach, we use a method of feature importance selection to reduce the number of features and keep only the most useful ones at the level of drowsiness detection for each individual (Figure 11). RFECV is the selection method employed in this work. 

RFECV provides us with the most important features for each subject and eliminates the least important. Table 4 presents the number of most important characteristics for each subject. Figure 12 represents The number of features selected by RFECV with SVM.

The research shows that the SVM with the radial basis function (RBF) kernel with a penalty factor C = 1 and kernel factor γ = 0.4 is the most exact classifier for detecting drowsiness in the intra mode, with just two C3–C4 derivations and seven features picked by RFECV, with an overall accuracy of 99.85%. Grid search optimization is used to select optimal hyperparameters. Figure 13 presents the accuracy for the train and test sets of SVM. 

### 6.2. Inter Mode

In this section, we will work with three temporal characteristics, five frequencies, and eight TFs, which gives us 16 characteristics per electrode. For two electrodes, we have 2 × 16 = 32 characteristics. Our work consists of evaluating the performance of our approach with two EEG derivations in four different data distribution protocols. Moreover, the results are compared with the intra mode to specify the most effective data distribution protocol to train a more generalist model that can eliminate and overcome the problem of EEG variability between subjects. 

Subsequently, we move on to the use of the RFECV to select the most important features and eliminate the less decisive features in connection with the detection of drowsiness. This feature selection method will help us decrease the features on the one hand and increase the system accuracy on the other hand.

As in the intra mode, we start by identifying the two electrodes that have the highest accuracy in the inter mode for later use (Figure 14).

Table 5, Table 6, Table 7, Table 8 and Table 9 present the classification results for the original data with only two EEG leads (C3 and C4) without the use of the feature significance selection method for four data distribution protocols.

Based on the results, we can see that the C3–C4 represents the most accurate leads in the inter mode, but we can also see that the accuracy of the approach considerably decreases. For this reason, the next phase is to use a significant feature selection method known as RFECV to select the most important features concerning drowsiness to increase the accuracy of the approach. Figure 15 shows the evolution of the precision of the approach according to the number of characteristics selected by RFECV.

The maximum accuracy obtained after the use of the RFECV in the inter mode is that of MLP with nine input neurons, one hidden layer containing 100 neurons, and two output neurons. The ReLU activation function is used for the hidden layer, and Adam is used as the optimization algorithm with a batch size of 64 and a value of 96.4% with protocol P4, as shown in Table 10, along with a number of characteristics selected by RFECV that are equal to nine features. Grid search optimization is used to select optimal hyperparameters. As a result, we can see that the accuracy of the approach considerably decreases with the reduction in the number of tracks as well as in the features compared to the results of the intra-subject mode. On the other hand, we can see that the approach can overcome EEG variability with a high accuracy rate. Figure 16 presents the accuracy of MLP in train and test phases. 

### 6.3. Comparison of RFECV with Other Feature Selection Methods

In this part, we compare the performance of the approach using other feature selection methods, K-PCA and PCA, to clarify the effect of the selection method on the performance of the approach. Figure 17 presents the results of the classification of the selection method proposed in this work with K-PCA and PCA with the data distribution of protocol P4.

Based on the results, we can clearly see that the RFECV is more effective as a feature selection method for drowsiness detection in comparison with K-PCA and PCA.

### 6.4. Discussion 

The main objective of this study is to propose a generalized EEG-based drowsiness detection approach using minimal EEG electrodes to ensure consistent accuracy across subjects. We extracted comprehensive EEG features (time, frequency, time–frequency) to capture drowsiness nuances. Intelligent feature selection via RFECV optimized relevant feature retention. Evaluation across intra and inter modes identified robust machine learning models for managing inter-subject variability while maintaining high detection accuracy.

This study can serve as a foundational step in developing a drowsiness monitoring device based on EEG signals. Numerous studies have indicated the potential of detecting drowsiness using EEG signals. 

To evaluate the detection precision and generalization of the approach across various subjects, we compare the performance of the ML models of the proposed approach using different databases. The comparison database (Sahloul University Hospital) [29] consists of EEG signals collected from eight healthy subjects aged 21 to 25 with no history of alcoholism or drug use. These EEG signals were recorded at the Vigilance and Sleep Center of the Faculty of Medicine in Monastir, following an experimental protocol approved by our faculty’s Ethics Committee. All participants signed an informed consent form, which included a brief description of the research involving human subjects, before starting the experiment. This database, containing 45 h of EEG data related to drowsiness, is currently available upon request from the concerned author. The data collection protocol required subjects to wake up before 10:00 AM and spend approximately four hours completing the task. Each subject’s record is represented by 19 EEG channels (Fp1, Fp2, F2, F3, Fz, F4, F8, T3, C3, Cz, C4, T4, T5, P3, Pz, P4, T6, O1, O2). 

The drowsiness detection results of the proposed approach using Sahloul University Hospital data [29] in the intra mode with SVM-RBF show an accuracy of 96.87%, precision of 95.9%, sensitivity of 97.2%, and an F1-score of 96.5%. For the inter mode, the MLP shows an accuracy of 92.5%, precision of 91.2%, sensitivity of 92.3%, and an F1-score of 93.1%. The inter-subject variability in the EEG data still impacts the results slightly. However, it is noteworthy that even when using diverse datasets, the approach maintains strong generalizability and precision in drowsiness detection. This resilience is attributed to the diversity of EEG features, robust feature selection methods, and appropriate classifier parameterization chosen for the analysis.

In Table 11, we compare our proposed method to existing methods in intra mode. In [22], the authors segmented the filtered EEG into 1 s intervals before extracting time–frequency (TF) features from 32 EEG channels using wavelet transformation, achieving 82% accuracy in drowsiness detection using KNN as a classifier. For feature selection, they employed the Neighborhood Component Analysis (NCA). In [24], the authors introduced an EEG-based approach to detect drowsiness, focusing only on spectral characteristics. They used fast Fourier transform (FFT) to extract frequency characteristics from 1 min EEG segments from four EEG channels and employed SVM-RBF for classification, resulting in 78% accuracy. Principal Component Analysis (PCA) was used for feature selection in this work. Another study in [25] used spectral characteristics (PSDs) extracted by FFT from 3 s EEG segments from 32 electrodes, reporting a 92.6% accuracy using a neural network (NN). In [26], using the same “DROZY” database as ours, the authors extracted Hjorth parameters from 2 s EEG segments from a single EEG channel (C3) and used MLP classifiers, achieving 90% accuracy. This work did not employ any feature selection method.

In Table 12, we compare our approach to other recent studies focusing on drowsiness detection in inter (cross-subject) mode. In [64], the authors utilized EEG entropy extracted from 1 s EEG segments for drowsiness detection in inter mode, employing a hybrid classifier (LR+ELM+LightGBM) to achieve 94% accuracy without using any feature selection method. In [65], researchers used 30 EEG channels to extract EEG spectral characteristics (relative power) from 3 s segments filtered with a bandpass filter [0,50] Hz and employed SVM as a classifier, achieving a 68.64% accuracy in drowsiness detection by using all the features without selection. In [38], EEG signals were filtered with a Butterworth low-pass filter and segmented into 1 s segments. EEG spectral characteristics were then extracted with FFT and deployed with a decision tree (DT) classifier, achieving an accuracy of 85.6% in inter mode drowsiness detection. In this work, Minimum Redundancy Maximum Relevance (MRMR) was used to reduce feature dimensions. Finally, in [66], the authors extracted power spectral density (PSD) from 32 filtered EEG channels segmented into 1 s segments and used SVM as a classifier, achieving an accuracy of 87.16% in drowsiness detection without using any feature selection methods.

We observe that our proposed approach demonstrates greater precision compared to the aforementioned works in both intra and inter modes, achieving accuracies of 99.85% and 96.4%, respectively, with only two electrodes, minimizing interference between electrodes. Additionally, the size of the EEG segments directly influences decision-making. The choice of 10 s EEG segments for drowsiness detection is justified by an optimal balance between temporal resolution and stability of the extracted features. Unlike shorter 1–3 s segments, which can be too noisy and variable, 10 s segments capture relevant EEG trends without losing granularity. Compared to 1 min segments, 10 s segments allow for a faster detection of drowsiness transitions while being long enough to ensure good representativeness of states. The quality of the EEG features also greatly influences the accuracy of the approach. Using different EEG features in various domains (time, frequency, TF) increases the system’s capacity for drowsiness detection, whereas using a single type of feature limits system capacity.

The choice of classification model is a critical phase that directly affects the performance of the approach. In the intra mode, SVM-RBF demonstrates a significant ability to handle the nonlinearity, complexity, and high dimensionality of EEG features for each subject. On the other hand, MLP minimizes and overcomes the effect of inter-subject EEG variability, enhancing the generalization of the approach. This precision in machine learning models is attributed to the careful selection of hyperparameters for each classifier, underscoring the importance of the hyperparameter optimization phase using grid search. The feature selection method is crucial for enhancing the precision of drowsiness detection. The influence of the selection method on detection accuracy is evident in the cited works. Selection methods based on dimension reduction techniques such as PCA and MRMR tend to be less accurate in detecting drowsiness compared to RFECV, which employs a machine learning model to retain the most relevant features. For instance, NCA uses KNN as a selection model, which can be limited due to the nonlinearity of EEG data. In contrast, SVM-RBF used with RFECV better captures the complex, nonlinear patterns in EEG signals, leading to higher precision in our proposed approach for both intra and inter modes.

This substantial difference in performance is mainly due to the following reasons:The different EEG features extracted from various analysis domains such as time, frequency, and time–frequency (TF) help increase the number of indicators of drowsiness, thereby enhancing the accuracy and generalization of the approach.The use of a 10 s sliding window helps maintain critical information about drowsiness, enabling more precise detection compared to a 30 s window. This choice significantly enhances the accuracy of drowsiness detection by capturing more immediate and relevant changes in the EEG signals. Consequently, the approach benefits from improved sensitivity to variations in drowsiness levels, resulting in a more reliable and effective monitoring system.The intelligent feature selection layer, composed of RFECV based on SVM-RBF, instead of dimension reduction tools like PCA and KPCA, helps maintain only the most relevant features related to drowsiness. Additionally, the K-fold cross-validation technique helps to eliminate overfitting, ensuring the model’s robustness and generalizability.The selection of suitable EEG channels (C3, C4) with the highest precision helps minimize the effect of interference between electrodes, enhancing the system’s accuracy and making it more adaptable to real-life conditions.

## 7. Conclusions

Decreased alertness, especially passive wakefulness (drowsiness), is a very dangerous condition in areas such as transportation, industry, and medicine. In this work, we have proposed an approach to drowsiness detection based on EEG features coming from two EEG leads (C3, C4). The suggested system uses the Welch method, EEG statistical characteristics, and the DWT to extract the different EEG characteristics in time, frequency, and TF domains, respectively. The RFE technique has been used as a selection method to keep the most important features to ensure more accurate and generic drowsiness detection. The different ML models have been utilized to differentiate two states of vigilance (awake, drowsy). The proposed system is capable of detecting drowsiness with an accuracy of 99.85% and 96.4%, respectively, in intra and inter modes. The strengths of the suggested approach are represented by their ability to overcome the inter-subject problem and they offer a more generalized system with a high accuracy rate. In addition, the proposed drowsiness detection system uses a limited number of EEG electrodes, which makes it more adaptable to real-life conditions. As a perspective, we aim to incorporate facial expressions as well as other physiological signals, such as EOG and ECG, with the EEG signal to strengthen the reliability of the approach. Additionally, testing the approach on various drowsiness-related databases is a crucial phase in evaluating the generalizability, robustness, and accuracy of this drowsiness detection method, reflecting our ongoing efforts to explore its potential across diverse datasets. Furthermore, the implementation of this approach on a programmable platform for the creation of an embedded drowsiness detection system represents a future topic to be studied.

## Figures and Tables

**Figure 1 sensors-24-04256-f001:**
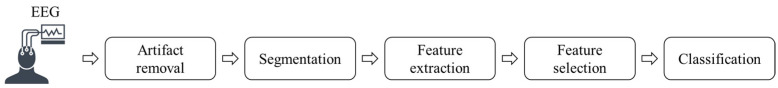
Drowsiness detection with EEG signals general processing chain.

**Figure 2 sensors-24-04256-f002:**
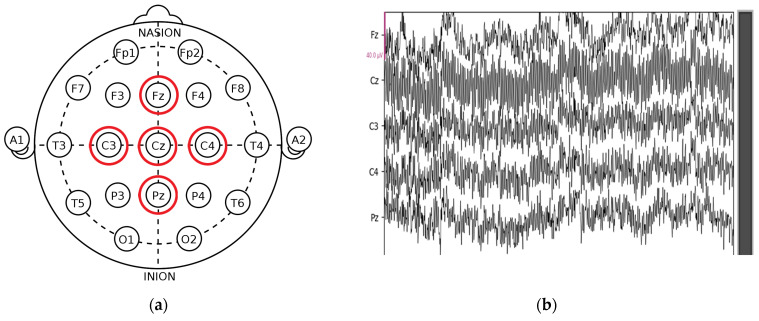
DROZY EEG signals [52]: (**a**) location of EEG electrodes according to international system 10–20 (Fz, Cz, C3, C4, Pz); (**b**) EEG raw.

**Figure 3 sensors-24-04256-f003:**
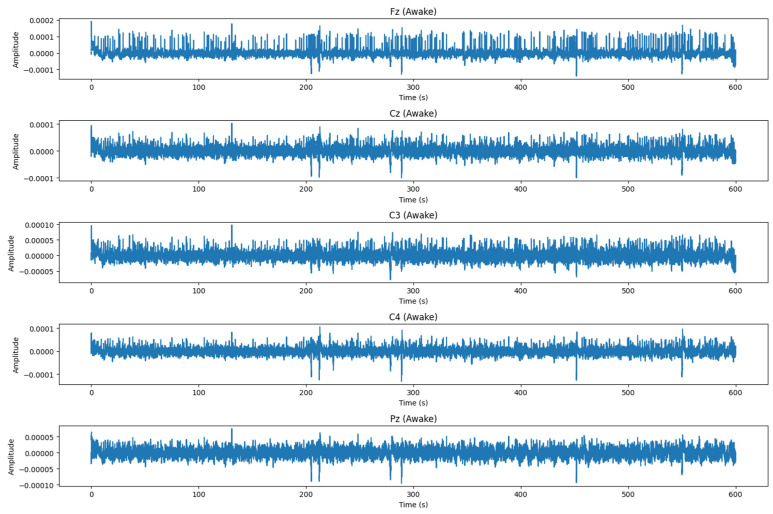
DROZY EEG signals for alert subject.

**Figure 4 sensors-24-04256-f004:**
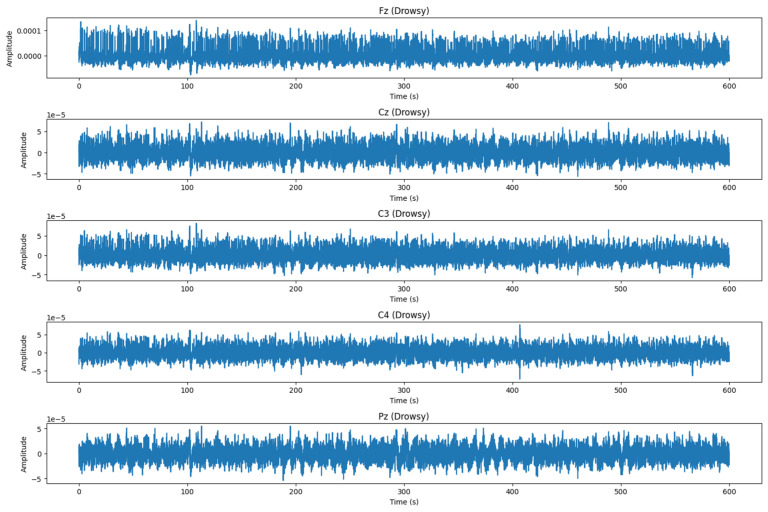
DROZY EEG signals for drowsy subject.

**Figure 5 sensors-24-04256-f005:**
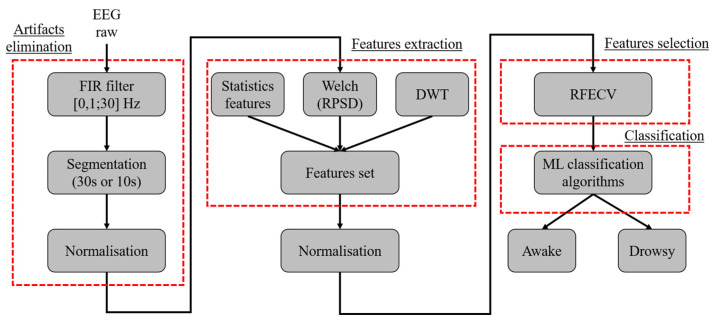
The general scheme of the proposed method.

**Figure 6 sensors-24-04256-f006:**
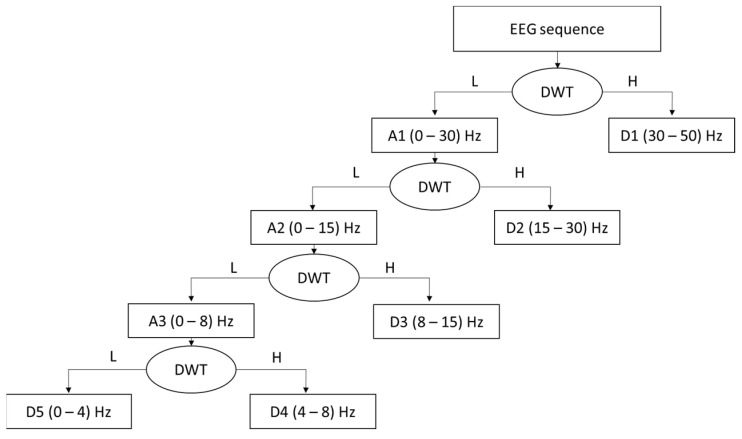
EEG signal decomposition by the wavelet function [39].

**Figure 7 sensors-24-04256-f007:**
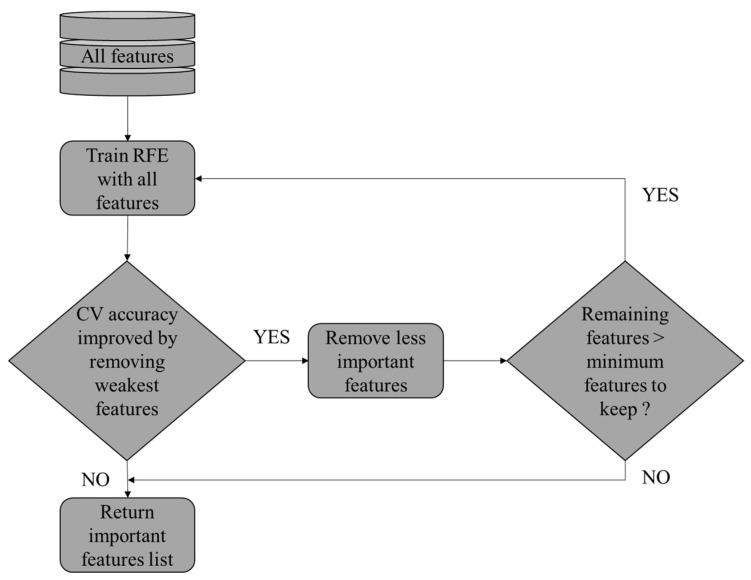
Principle of operation of the RFECV.

**Figure 8 sensors-24-04256-f008:**
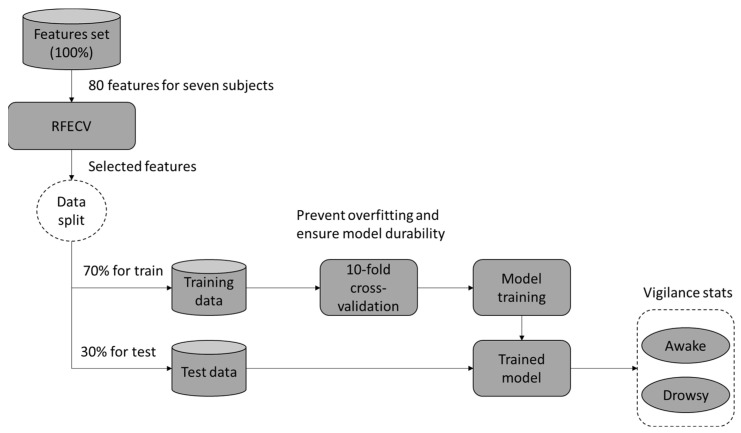
Data distribution for the train and test sets in the intra mode.

**Figure 9 sensors-24-04256-f009:**
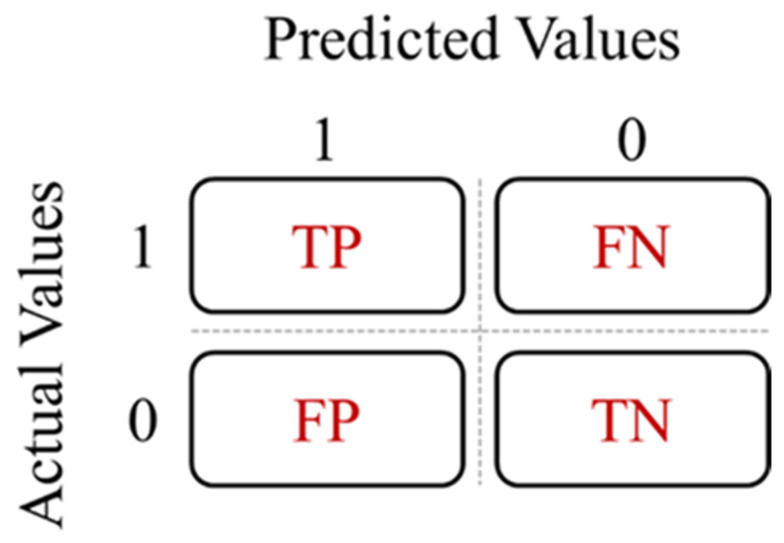
Confusion matrix. True Positive (TP): prediction of drowsiness when the actual state is drowsiness. False Positive (FP): prediction of drowsiness when the real state is alertness. True Negative (TN): prediction of alertness when the real state is alertness. False Negative (FN): prediction of alertness when the real state is drowsiness.

**Figure 10 sensors-24-04256-f010:**
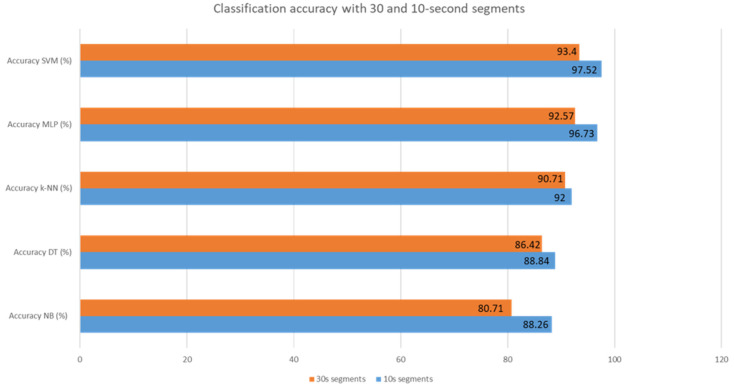
Classification accuracy with 30 and 10 s segments.

**Figure 11 sensors-24-04256-f011:**
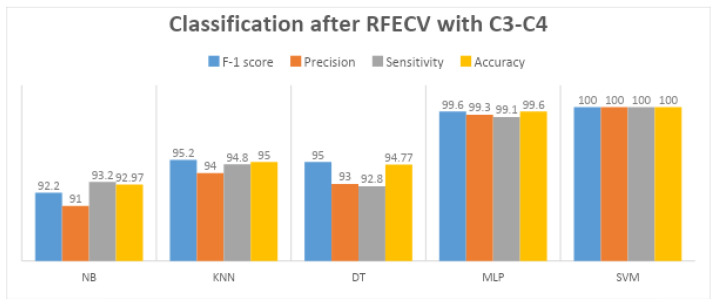
Different classifiers’ accuracy after RFECV.

**Figure 12 sensors-24-04256-f012:**
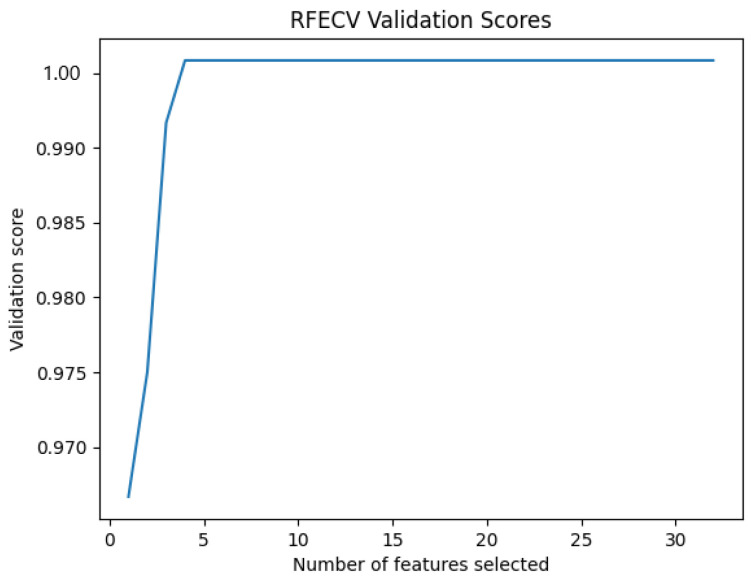
The number of features selected by RFECV with SVM.

**Figure 13 sensors-24-04256-f013:**
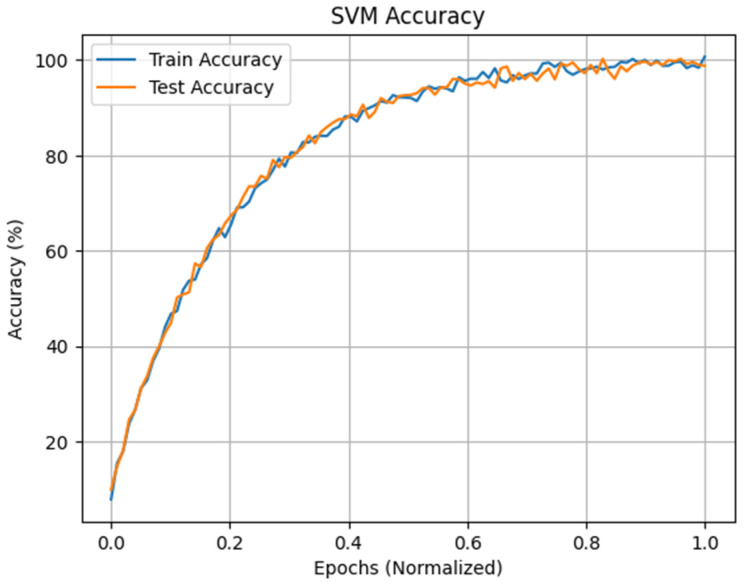
SVM train and test accuracy.

**Figure 14 sensors-24-04256-f014:**
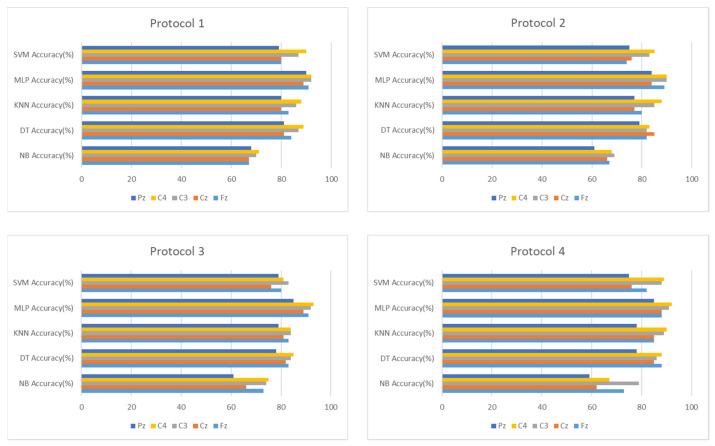
Different classifiers’ accuracy with different EEG deviations in the inter mode.

**Figure 15 sensors-24-04256-f015:**
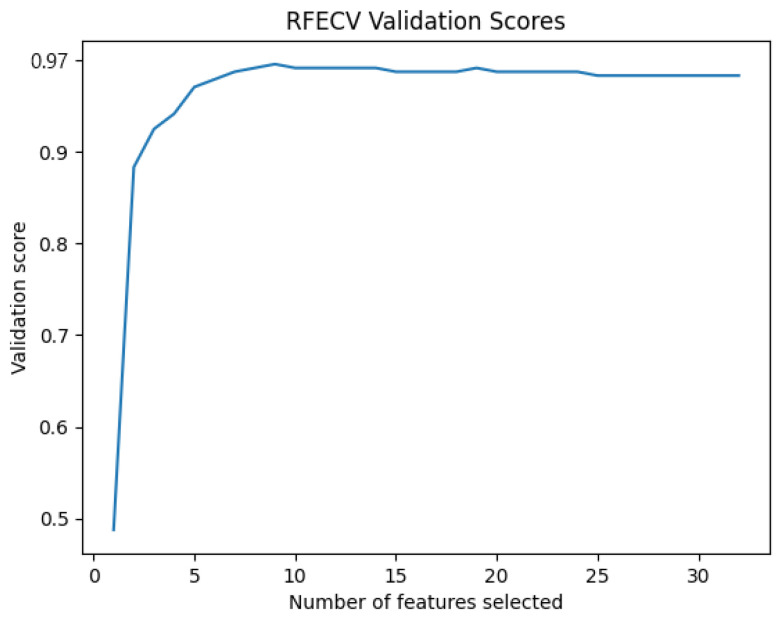
The number of features selected by RFECV with MLP.

**Figure 16 sensors-24-04256-f016:**
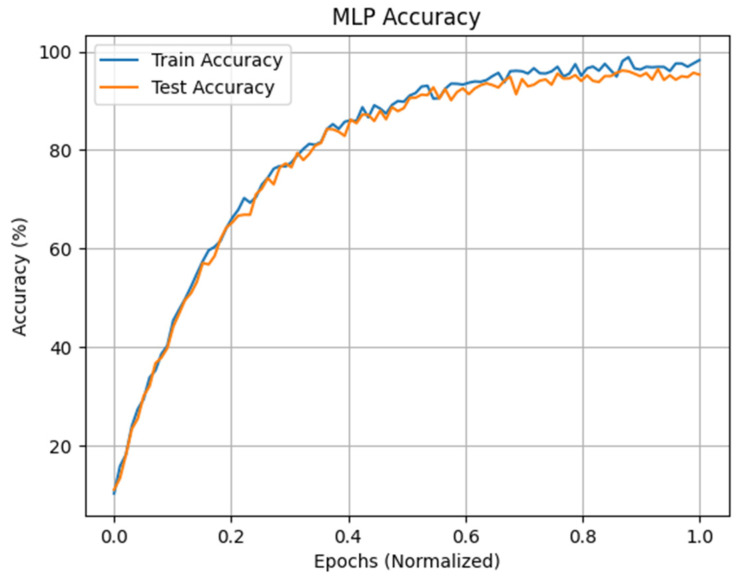
MLP train and test accuracy.

**Figure 17 sensors-24-04256-f017:**
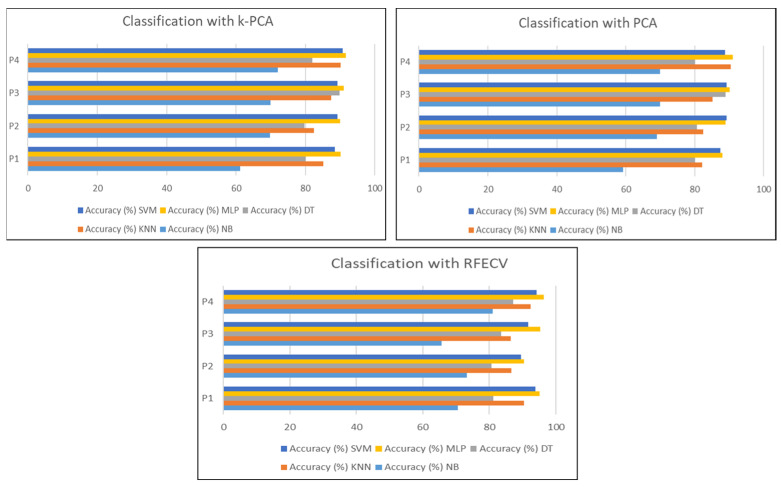
The effect of the feature selection method on the accuracy of the approach.

**Table 1 sensors-24-04256-t001:** Data distribution protocols in the inter mode.

Protocol Names	Train Data	Test Data
P1 (combined subject)	70% of all features	30% of all features
P2 (cross-subject)	Six subjects	One subject
P3 (cross-subject)	Five subjects	Two subjects
P4 (cross-subject)	Four subjects	Three subjects

**Table 2 sensors-24-04256-t002:** Different classifiers’ accuracy with 10 s segments in the intra mode.

Subjects	NB (Accuracy %)	KNN (Accuracy %)	DT (Accuracy %)	MLP (Accuracy %)	SVM (Accuracy %)
Subject 1	78	81.9	82	94	95.8
Subject 2	81	86	81	94.4	98
Subject 3	87.5	94.4	88.8	99	98.6
Subject 4	99.6	98.95	95.8	99.9	99
Subject 5	84.72	87.5	95.6	97.2	97.5
Subject 6	94	94.4	94	98.6	98.8
Subject 7	93	86	84.7	94	94
Overall	88.26	89.87	88.84	96.72	97.38

**Table 3 sensors-24-04256-t003:** Different classifiers accuracy with different EEG deviation in the intra mode.

Derivation	NB (Accuracy %)	DT (Accuracy %)	KNN (Accuracy %)	MLP (Accuracy %)	SVM (Accuracy %)
Fz	82.8	67.52	79.92	71.2	75.5
Cz	83	79.2	79.1	75	83.5
C3	87.8	88.1	85.6	90.2	91.8
C4	88.5	88.8	83.2	92.1	94.8
Pz	62.2	65	75.8	80.2	83

**Table 4 sensors-24-04256-t004:** The number of the important EEG features selected by RFECV for each subject.

Subjects	Number of Features	Name of Features
S1	7	Skewness (C3)/Standard deviation of details coefficients (c4)/Delta RPSD (c3)/Beta RPSD (c3)/Beta RPSD (c4)/Gamma RPSD (c3)/Gamma RPSD (c4)
S2	9	Standard deviation (c4)/Kurtosis (c4)/Energy of details coefficients (c4)/Theta RPSD (c4)/Alpha RPSD (c4)/Beta RPSD (c3)/Beta RPSD (c4)/Gamma RPSD (c3)/Gamma RPSD (c4)
S3	7	Standard deviation (c4)/Kurtosis (c4)/Standard deviation of details coefficients (c4)/Alpha RPSD (c4)/Beta RPSD (c4)/Beta RPSD (c3)/Gamma RPSD(c4)
S4	4	Delta RPSD (c4)/Theta RPSD (c4)/Beta RPSD (c3)/Gamma RPSD (c4)
S5	8	Standard deviation (c3)/Standard deviation (c4)/Skewness (C3)/Skewness (C4)/Kurtosis(c4)/Energy of details coefficients (c3)/Energy of details coefficients (c4)/Energy of approximation coefficients (c4)
S6	9	Energy of details coefficients (c3)/Energy of details coefficients (c4)/Energy of approximation coefficients (c4)/Energy of approximation coefficients (c3)/Entropy of details coefficients (c4)/standard deviation (c4)/Skewness (C3)/Mean of details coefficients (c4)/standard deviation of approximation coefficients (c4)
S7	19	Entropy of details coefficients (c4)/Entropy of details coefficients (c3)/Energy of details coefficients (c3)/Energy of details coefficients (c4)/Energy of approximation coefficients (c4)/Energy of approximation coefficients (c3)/Skewness (C3)/Skewness (C4)/Theta RPSD (c3)/Alpha RPSD (c4)/Alpha RPSD (c3)/Beta RPSD (c3)/Beta RPSD (c4)/Gamma RPSD (c3)/Gamma RPSD (c4)/Standard deviation (c4)/Kurtosis(c4)/Standard deviation (c3)/Kurtosis(c3)

**Table 5 sensors-24-04256-t005:** NB accuracy with only C3 and C4.

Protocols	NB			
	P (%)	S (%)	F1 (%)	A (%)
P1	66.1	65.2	65	65.7
P2	71.5	71.1	72.1	71.2
P3	63.1	61.5	62.8	62.65
P4	79	77.8	78.5	78.2

**Table 6 sensors-24-04256-t006:** KNN accuracy with only C3 and C4.

Protocols	KNN			
	P (%)	S (%)	F1 (%)	A (%)
P1	86.5	84.8	85.6	85.2
P2	84.5	84.5	85.2	84.63
P3	84.8	84.3	87.1	85.5
P4	88.1	87.5	89.1	88.3

**Table 7 sensors-24-04256-t007:** DT accuracy with only C3 and C4.

Protocols	DT			
	P (%)	S (%)	F1 (%)	A (%)
P1	78.1	78.7	79.9	79.5
P2	79.5	78.2	80.9	79.7
P3	82.3	82.1	83	81.89
P4	86.3	85	86.1	85.2

**Table 8 sensors-24-04256-t008:** MLP accuracy with only C3 and C4.

Protocols	MLP			
	P (%)	S (%)	F1 (%)	A (%)
P1	92.5	94	93.2	93.8
P2	88.7	88.5	90	88.99
P3	94	94	94	94
P4	92.9	94.8	95.2	94.8

**Table 9 sensors-24-04256-t009:** SVM accuracy with only C3 and C4.

Protocols	SVM			
	P (%)	S (%)	F1 (%)	A (%)
P1	88	88	88.1	88
P2	81.3	80.5	79.89	80.2
P3	85.5	84.2	86.2	85.3
P4	89.5	89.1	89.7	88.97

**Table 10 sensors-24-04256-t010:** Different classifiers’ accuracy after RFECV in the inter mode.

Protocols	NB (Accuracy %)	DT (Accuracy %)	KNN (Accuracy %)	MLP (Accuracy %)	SVM (Accuracy %)
P1	70.6	90.5	81.2	95.18	93.85
P2	73.2	86.63	80.7	90.5	89.51
P3	65.65	86.5	83.5	95.3	91.8
P4	81	92.4	87.2	96.4	95.2

**Table 11 sensors-24-04256-t011:** Comparative analysis of the proposed method versus other systems in intra mode.

Ref.	Feature Extraction Method	Classifier	Database	Number of Electrodes	A (%)	P (%)	S (%)	F1 (%)
[22]	WT	KNN	Private	32	82.08	78.84	87.71	83.27
[24]	FFT	SVM	Private	4	78.3	80.92	78.95	76.51
[25]	PSD	Neural network	EEG driver drowsiness dataset [63]	32	92.6	92.7	-	92.7
[26]	Hjorth Parameters	MLP	DROZY	1	90	-	-	-
[28]	PSD	SVM	DROZY	5	96.4			
[30]	TQWT	SVM	Sahloul University Hopital	1	94	-	94.08	-
Proposed methods	Statics/RPSD/ DWT	SVM	DROZY	2	99.85	99.87	99.8	99.5

**Table 12 sensors-24-04256-t012:** Comparative analysis of the proposed method versus other systems in inter mode.

Ref.	Feature Extraction Method	Classifier	Database	Number of Electrodes	A (%)	P (%)	S (%)	F1 (%)
[64]	Entropy	Hybrid classifier (LR+ELM+LightGBM)	Private	2	94.2	-	94	-
[65]	Relative power	SVM	Multichannel_EEG_recordings_during_a_sustainedattention_driving_task_preprocessed_dataset	30	68.64	-	-	-
[38]	Spectral power	DT	Private (HITEC University, Taxila, Pakistan)	1	85.6	89.7	-	87.6
[66]	PSD	SVM	Private (North eastern UNIVERSITY)	32	87.16	-	-	-
[29]	TQWT	SVM	Private (Sahloul University Hopital)	1	89	-	89.37	-
Proposed methods	Statics/RPSD/DWT	SVM	DROZY	2	96.4	96.9	95.87	96

## Data Availability

The original data presented in the study are openly available [DOI:10.20944/preprints202405.1615.v1].

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
