# Peer review of "Efficient Generalized Electroencephalography-Based Drowsiness Detection Approach with Minimal Electrodes"

_sensors, 2024, doi:10.3390/s24134256_

Round 1

Reviewer 1 Report

Comments and Suggestions for Authors

This research proposes an Electroencephalography (EEG) based approach for detecting drowsiness. EEG signals are passed through a preprocessing chain composed of artifact removal and segmentation to ensure accurate detection followed by different feature extraction methods to extract the different features related to drowsiness. Specific comments are given as follows.

1. The structural details of each machine learning model are unclear, such as SVM, KNN, NB, DT, and MLP. The technical details of the model should be supplemented, such as the size and step size of the parameters.

2. The specific feature selection and extraction (such as RFECV) should be explained in detail.

3. In Figure 6, the training details and process should be supplemented.

4. The optimization process of the objective function of each machine learning model needs to be demonstrated to help observe its convergence.

5. The introduction of Discrete Wavelet Transformation is confusing. Which kind of wavelet function is used for DWT?

6. More experiments should be added, including comparisons and ablations.

7. Some examples of EEG signals are encouraged to be shown.

8. Related work in the field is insufficient. Some important related work should be cited and discussed, such as MR-DCAE: Manifold regularization-based deep convolutional autoencoder for unauthorized broadcasting identification, An adaptive global–local generalized FEM for multiscale advection–diffusion problems, Machine learning-based prediction models for patients no-show in online outpatient appointments.

Author Response

Manuscript ID: sensors-3033148
Type of manuscript: Article
Title: Efficient Generalized EEG-based Drowsiness Detection Approach with
Minimal Electrodes
Authors: Aymen Zayed *, Nidhameddine Belhadj, Khaled Ben Khalifa, Mohamed
Hedi Bedoui, Carlos Valderrama
Submitted: 13 May 2024

Abstract: Drowsiness is a main factor for various costly defects, even fatal accidents in areas such as construction, transportation, industry and medicine, due to the lack of monitoring vigilance in the mentioned areas. The implementation of a drowsiness detection system can greatly help to reduce the defects and accident rates by alerting individuals when they enter a drowsy state. This research proposes an Electroencephalography (EEG) based approach for detecting drowsiness. EEG signals are passed through a preprocessing chain composed of artifact removal and segmentation to ensure accurate detection followed by different feature extraction methods to extract the different features related to drowsiness. This work explores the use of various machine learning algorithms such as Support Vector Machine (SVM) the K Nearest Neighbor (KNN) the Naive Bayes (NB) the Decision Tree (DT) and the Multilayer Perceptron (MLP) to analyze EEG signals sourced from the DROZY database, carefully labeled into two distinct states of alertness (awake, and drowsy). Segmentation into 10-second intervals ensures precise detection, while a relevant feature selection layer enhances accuracy and generalizability. The proposed approach achieves high accuracy rates of 99.84% and 96.4% for intra (subject by subject) and inter (cross-subject) modes, respectively. SVM emerges as the most effective model for drowsiness detection in the intra mode, while MLP demonstrates superior accuracy in the inter mode. This research offers a promising avenue for implementing proactive drowsiness detection systems to enhance occupational safety across various industries.

We are very thankful to the reviewers for their deep and interesting reviews. We have revised our research paper in light of their useful suggestions and comments.

Response to reviewers

Reviewer 1: This research proposes an Electroencephalography (EEG) based approach for detecting drowsiness. EEG signals are passed through a preprocessing chain composed of artifact removal and segmentation to ensure accurate detection followed by different feature extraction methods to extract the different features related to drowsiness. Specific comments are given as follows.

Comment 1

The structural details of each machine learning model are unclear, such as SVM, KNN, NB, DT, and MLP. The technical details of the model should be supplemented, such as the size and step size of the parameters.

Response

Thank you for your valuable feedback regarding the structural details of the machine learning models employed in our study. We recognize the importance of providing comprehensive information on model parameters to enhance reproducibility and facilitate a deeper understanding of our methodology. Technical details of each model are supplemented as follows:

  1. A comprehensive description of the Support Vector Machine (SVM) model, with a specific focus on the Radial Basis Function (RBF) kernel and its associated hyperparameters, has been added to Section 5.2 on pages 15 and 16. This includes detailed explanations of the equation of the hyperplane and the classification operation, aimed at enhancing understanding and clarity for readers. The modifications are presented in the text below :

“SVMs [52] are ML algorithms used in machine learning to solve problems of classification, regression, or anomaly detection. 

The main goal of the SVM is to find a hyperplane that separates the different classes with the widest possible margin. Equation (15) define the SVM hyperplane equation:

                                                                                       (15)                                                               

With  represents the weight vector, b is the bias and x is the feature vector.

For optimal separation, SVM maximizes the margin  by synchronizing  under the constraint represented by the following equations (16) (17):

                                                                                      (16)

                           with                                  (17)

Where (xi,yi) represents the training dataset, which is 1<i<N. x represents the characteristic vector extracted from the EEG signals, y indicates the corresponding vigilance status labels, and N is the number of data. Moreover, K is the kernel function of the SVM, Si is the vector support,  are the weights, and b is the bias. The SVM algorithm supports more than one kernel (linear, polynomial, RBF, sigmoid), and the choice of kernel always depends on the data type. Equation (18) represents SVM with the radial basis function (RBF) kernel:

                                                                          (18)

This kernel adeptly handles non-linear EEG data, capturing intricate brain activity patterns associated with drowsiness. The penalty factor C and the kernel factor  represent the SVM-RBF hyperparameters.

  1. The equation of the K-Nearest Neighbors (KNN) classification operation has been added in Section 5.2 on page 16. These additions aim to enhance understanding and clarity for readers. The modifications are presented in the text below :

“ For the KNN classification operation, the new sample x is assigned to the majority class among its k nearest neighbors. The equation (20) represents the predicted class :

                                                                       (20)

                     Where  represents the set of the k nearest neighbors of x. “

  1. A clear description of the Naive Bayes (NB) model has been incorporated into Section 5.2 on pages 16 and 17. This includes an explanation of the Bayes theorem, as well as a detailed description of the model's parameters and the classification operation. These additions aim to improve understanding and clarity for readers. The modifications are presented in the text below :

“ The following equation () represents the Bayes theorem:

                                                                                   ()

Where P(y|x) represents the probability of class y given features x, P(x|y) is the probability of features x given class y, P(y) is the probability of class y, and P(x) is the probability of features x.

For the classification operation, the NB algorithm calculates the probability of each class and chooses the one with the highest probability as presented by equation ():

                                           “                                     

  1. An explanation of the classification operation and parameters for the Decision Tree (DT) model in Section 5.2 on page 17. These additions provide readers with a clearer understanding of how the DT model works and the significance of its parameters in the classification process. The modifications are presented in the text below :

“ For classification, the DT algorithm is based on impurity measurements, such as entropy or the Gini index. Entropy is given by equation (23) and the Gini index by equation (24):

                                                                        (23)

                                                                                (24)

)

 “

  1. we have supplemented Section 5.2 on pages 17 and 18 with an explanation of the classification operation and the parameters of the Multilayer Perceptron (MLP) model. Additionally, we have provided insights into the adaptability of MLP with EEG drowsiness detection, highlighting its efficacy in capturing complex patterns and non-linear relationships within EEG data. The modifications are presented in the text below :

“ The equation (25) presents the output layer:

                                                                              (25)

 is the weight of the connection between neuron  and neuron ,  is the input ,  is the bias of neuron j,  is the activation function.

The loss function for classification with c classes is often cross-entropy (26):

                                                                           (26)

Alternatively,  is the binary indicator if the example  belongs to the class .

 is the predicted probability that the example  belongs to the class .

MLPs can directly handle high-dimensional EEG data, learning complex relationships between features and drowsiness levels. They offer flexibility in modeling brain activity patterns, capable of capturing both linear and nonlinear relationships. However, careful hyperparameter tuning and regularization are essential to prevent overfitting and ensure robust performance in EEG drowsiness detection tasks.

Additionally, to ensure the selection of optimal parameters for our models, we employed the Grid Search technique. Grid Search systematically explores a range of hyperparameter values for each model and selects the combination that yields the best performance based on specified evaluation metrics. This approach enhances the robustness of our model tuning process by exhaustively searching through the parameter space. The values of the hyperparameters for the most accurate models in both INTRA (SVM) and INTER (MLP) modes, selected using Grid Search, have been included in

Section 6.1 on page 21 in the text below :

”SVM with the Radial Basis Function (RBF) kernel with penalty factor C=1 and kernel factor =0.4”

 Section 6.2 on page 24 in the text below :

” MLP with nine input neurons, one hidden layer containing 100 neurons, and two output neurons. The ReLU activation function is used for the hidden layer, and Adam is used as the optimization algorithm with a batch size of 64 ” 

Comment2

The specific feature selection and extraction (such as RFECV) should be explained in detail.

Response

  • Feature selection details have been added in section 4.2, page 13, The modifications are presented in the text below :

“ RFECV represents a sophisticated approach to feature selection that merges the advantages of recursive features elimination (RFE) that represents a feature selection method that recursively eliminates the least important features and builds the model with the remaining features [17] with cross-validation (CV) which represents a technique used to evaluate model performance by partitioning data into training and testing sets multiple times. This technique is especially useful when the goal is to choose the most relevant features for a ML model, while simultaneously estimating the optimal number of features to consider. RFECV is distinguished by its focus on automating this complex process and determining the ideal number of features to maximize model performance.

The RFECV [51] starts with an initial ML model known as M and a complete set of features called X. After evaluating the contribution of each feature to model performance. It iteratively removes the least important ones. Following each elimination, cross-validation as K-fold is used to assess the performance of the model. Equation (13) represents the cross-validation score ():

                                                                  (13)

X (train i) and X (test i) are the training and testing sets in each fold.

This process repeats until a predefined criterion, such as model accuracy, reaches an optimum or an optimal number of features  is identified. The selected feature set  with the highest cross-validation score is represented in Equation (14):

                                                                            (14)

Cross-validation is crucial in the RFECV process as it ensures that feature selection is robust and generalizable. By partitioning the data into subsets, cross-validation evaluates the performance of the model across various datasets, thus reducing the risk of over-fitting. The choice of the initial model M is based on the nature of the input data. For EEG features the SVM with the Radial Base Function (RBF) kernel represents the most used model because of these capabilities to handle the nonlinearity of EEG data [28]. “

  • The explanation of the choice of feature extraction tools has been added in section 4.1, page 10, in the text below ::

“The choice of EEG features is indeed crucial for effective drowsiness detection, as each analysis domain offers unique insights into indicators of drowsiness. In this study, we focused on extracting EEG features from different domains time, frequency, and time-frequency using robust tools such as statistical features over time, the Welch method, and Discrete Wavelet Transform. These techniques were selected to enhance the sensitivity and the generalizability of our drowsiness detection approach [11].”

Comment3

In Figure 6, the training details and process should be supplemented.

Response

Thank you for your comment. In response, we have enhanced Figure 6 by supplementing it with additional training details and processes. This update aims to provide a more comprehensive depiction of our methodology and ensure clarity for the readers.

The reference of Figure 6 is changed to Figure 8.

Comment4

The optimization process of the objective function of each machine learning model needs to be demonstrated to help observe its convergence.

Response

  • The details of the optimization process for the objective function of the selected machine learning model in INTRA mode have been added in section 6.1, page 21, as follows in the text below:

“SVM with the Radial Basis Function (RBF) kernel with penalty factor C=1 and kernel factor =0.4 is the most exact classifier for detecting drowsiness in the intra mode, with just two C3-C4 derivations and seven features picked by RFECV, with an overall accuracy of 99.85%. Grid search optimization is used to select optimal hyperparameters. Figure 13 represents the accuracy for the train and test sets of SVM. “

Accuracy curves for both training and test sets are clearly depicted in Figure 13 to illustrate the convergence of the model.

  • The details of the optimization process for the objective function of the selected machine learning model in INTER mode have been added in section 6.2, page 24, as follows in the text below :

“MLP with nine input neurons, one hidden layer containing 100 neurons, and two output neurons. The ReLU activation function is used for the hidden layer, and Adam is used as the optimization algorithm with a batch size of 64 with a value of 96.4% with protocol P4 as shown in Table 10 with a number of characteristics selected by RFECV equal to nine features. Grid search optimization is used to select optimal hyperparameters. As a result, we can see that the accuracy of the approach considerably decreases with the reduction in the number of tracks as well as in the features compared to the results of the intra-subject mode. On the other hand, we can see that the approach can overcome EEG variability with a high accuracy rate. Figure 16 represents the accuracy of MLP in train and test phases. “

To complement this description, corresponding accuracy curves are presented in Figure 16 to facilitate a precise visualization of the model's performance.

We hope these details meet your expectations and clearly illustrate our machine learning model optimization process.

Comment5

The introduction of Discrete Wavelet Transformation is confusing. Which kind of wavelet function is used for DWT?

Response

The introduction of the Discrete Wavelet Transformation has been revised in Section 4.1 page 12, as follows in the text below:

“DWT is a powerful mathematical tool used in many fields such as signal processing to perform many tasks. For EEG signals processing, DWT represents one of the most accurate tools in terms of feature extraction that guarantees the time-frequency analysis of the EEG signal. The working principle of the DWT consists in extracting two types of coefficients that ensure an accurate analysis of EEG signals in the temporal frequency domain. “

Regarding the type of wavelet used in this work, we employed the Daubechies wavelet function ("db4") for the extraction of both detail and approximation coefficients as mentioned in section 4.1 page 12:

“Our approach employs the Daubechies wavelet function (“db4”) for coefficient extraction,”

Comment6

More experiments should be added, including comparisons and ablations.

Response

More experiments have been added and discassed in the comparisons for the intra and inter mode, as detailed in Section 6.4, Table 12, on page 26.

Ref

Feature extraction method

Classifier

Database

Electrodes

number

A(%)

P(%)

S(%)

F1(%)

[61]

Entropy

Hybrid classifier (LR+ELM+LightGBM)

Private

2

94.2

-

94

-

[62]

Relative Power

SVM

Multichannel_EEG_recordings_during_a_sustainedattention_driving_task_preprocessed_dataset

30

68.64

-

-

-

[63]

Spectral power

DT

Private

(HITEC University, Taxila, Pakistan)

1

85.6

89.7

-

87.6

[64]

PSD

SVM

Private

(North eastern UNIVERSITY)

32

87.16

-

-

-

[28]

TQWT

SVM

Sahloul University Hopital

1

89

-

89.37

-

Proposed methods

Statics / RPSD /

DWT

SVM

DROZY

2

96.4

96.9

95.87

96

Other recent works have been added to Table 12 to enrich the comparison in INTER mode. The references are presented in the text below :

[61]     MIN, Jianliang, XIONG, Chen, ZHANG, Yonggang, et al. Driver fatigue detection based on prefrontal EEG using mul-ti-entropy measures and hybrid model. Biomedical Signal Processing and Control, 2021, vol. 69, p. 102857.

[62]     CUI, Jian, LAN, Zirui, SOURINA, Olga, et al. EEG-based cross-subject driver drowsiness recognition with an interpretable convolutional neural network. IEEE Transactions on Neural Networks and Learning Systems, 2022.

[63]     ARIF, Saad, MUNAWAR, Saba, et ALI, Hashim. Driving drowsiness detection using spectral signatures of EEG-based neu-rophysiology. Frontiers in physiology, 2023, vol. 14, p. 1153268.

[64]     WANG, Fei, WU, Shichao, PING, Jingyu, et al. EEG driving fatigue detection with PDC-based brain functional network. IEEE Sensors Journal, 2021, vol. 21, no 9, p. 10811-10823.

Comment7

Some examples of EEG signals are encouraged to be shown.

Response

We have added two new figures illustrating EEG signals for both alert and drowsy subjects in section 3.2 page 9.

Figure 3. presents an example of five EEG canals from an alert subject.

Figure 4. presents an example of five EEG canals from a drowsy subject.

Comment8

Related work in the field is insufficient. Some important related work should be cited and discussed, such as MR-DCAE: Manifold regularization-based deep convolutional autoencoder for unauthorized broadcasting identification, An adaptive global–local generalized FEM for multiscale advection–diffusion problems, Machine learning-based prediction models for patients no-show in online outpatient appointments.

Response

References addressing the suggested related work, including MR-DCAE, adaptive global–local generalized FEM, and machine learning-based prediction models for patient no-show in outpatient appointments, have been added in the Related Work section on pages 3 and 4 as follows:

”In addition, artificial intelligence has been seamlessly integrated across various domains, playing a pivotal role in executing numerous tasks, as evidenced by instances such as prediction models based on machine learning, notably those developed by Guorui Fan et al. [58], which represent a promising solution to anticipate patients' absences from online outpatient appointments. Using various data such as age, gender, and medical history, these models identify predictive trends. Different algorithms are evaluated, such as logistic regression, random forests, SVMs, and neural networks. The results show that neural networks and random forests outperform others in terms of accuracy, recall, accuracy, and F1-score. With up to 90% accuracy, these models offer significant potential to improve the operational efficiency of healthcare facilities and reduce absenteeism.

The adaptive global–local generalized FEM for multiscale advection–diffusion problems, proposed by Lishen He et al. [59], addresses multiscale advection-diffusion problems by combining adaptivity with global-local techniques. It employs generalized finite elements to integrate enrichment functions, capturing fine features on coarse meshes. Adaptivity adjusts the mesh and enrichment functions locally based on estimated errors, thus optimizing computational efficiency. The global-local approach solves the problem globally while focusing locally on areas with high gradients. Results show a significant improvement in precision, reducing error from 12% to 3%, and a reduction in computational cost, with a 50% decrease in computation time compared to traditional methods. This approach offers an effective and precise solution for complex multiscale problems.

Qinghe Zheng et al. [60] proposed a method called MR-DCAE (Manifold Regularization-Based Deep Convolutional Autoencoder) aimed at identifying unauthorized broadcasts using deep convolutional autoencoders combined with manifold regularization. This model extracts essential features from input data, ensures faithful reconstruction, and preserves the geometric structure of the data through regularization. By training the model on preprocessed data, it offers a compact and informative representation for identification tasks. Results show a precision of 95.6%, a recall rate of 94.3%, and an F1-Score of 94.9%, demonstrating high efficiency and robustness. This system is particularly suitable for real-time monitoring, intellectual property protection, and legal enforcement against illegal broadcasts.

For all these reasons, the integration of machine learning algorithms into the approach for detecting drowsiness represents a crucial advancement.”

[58]      FAN, Guorui, DENG, Zhaohua, YE, Qing, et al. Machine learning-based prediction models for patients no-show in online outpatient appointments. Data Science and Management, 2021, vol. 2, p. 45-52.

[59]      HE, Lishen, VALOCCHI, Albert J., et DUARTE, Carlos Armando. An adaptive global–local generalized FEM for multiscale advection–diffusion problems. Computer Methods in Applied Mechanics and Engineering, 2024, vol. 418, p. 116548.

[60]      ZHENG, Qinghe, ZHAO, Penghui, ZHANG, Deliang, et al. MR‐DCAE: Manifold regularization‐based deep convolutional autoencoder for unauthorized broadcasting identification. International Journal of Intelligent Systems, 2021, vol. 36, no 12, p. 7204-7238.

Reviewer 2 Report

Comments and Suggestions for Authors

The paper suffers from a significant lack of detailed information. Its discussion is too shallow.

I think the paper should be extensively modified.

In the title, it’s written as ‘APproach’.

The authors filtered raw signals at 30 Hz. How can you say higher frequency components are irrelevant? While it is true that high frequency components have higher risks of containing artifacts, it doesn’t mean there are no relevant components in such high frequencies. The authors should justify this choice.

The authors should define acronyms at their first appearance before using it (RPSD). In addition, the window size for welch’s method was not disclosed.

For the 5.2,

The authors must disclose all parameters for each method. (For example, the authors did not even mention what kernel they used for SVM. )

There are two ‘5.1’.

I think the authors also know that several ML algorithms’ performance depends on a lot of factors. So, just comparing them to conclude SVM is the best is barely able to generalize.

The authors also did not explain the use of features. What motivated the use of these features? For better generalization, a deliberate selection process is needed for features. Just putting everything and performing several algorithms without rationale for choosing the hyperparameters or something are likely to induce overfitting.

In the same vein, to me, it looks like the authors just used the channels simply because they were available in the dataset.

What is the core finding of the study?

I cannot find a real discussion. The current 6.4 is just a repetition of results. I strongly suggest inserting an additional section for a real discussion.

Author Response

Manuscript ID: sensors-3033148
Type of manuscript: Article
Title: Efficient Generalized EEG-based Drowsiness Detection Approach with
Minimal Electrodes
Authors: Aymen Zayed *, Nidhameddine Belhadj, Khaled Ben Khalifa, Mohamed
Hedi Bedoui, Carlos Valderrama
Submitted: 13 May 2024

Abstract: Drowsiness is a main factor for various costly defects, even fatal accidents in areas such as construction, transportation, industry and medicine, due to the lack of monitoring vigilance in the mentioned areas. The implementation of a drowsiness detection system can greatly help to reduce the defects and accident rates by alerting individuals when they enter a drowsy state. This research proposes an Electroencephalography (EEG) based approach for detecting drowsiness. EEG signals are passed through a preprocessing chain composed of artifact removal and segmentation to ensure accurate detection followed by different feature extraction methods to extract the different features related to drowsiness. This work explores the use of various machine learning algorithms such as Support Vector Machine (SVM) the K Nearest Neighbor (KNN) the Naive Bayes (NB) the Decision Tree (DT) and the Multilayer Perceptron (MLP) to analyze EEG signals sourced from the DROZY database, carefully labeled into two distinct states of alertness (awake, and drowsy). Segmentation into 10-second intervals ensures precise detection, while a relevant feature selection layer enhances accuracy and generalizability. The proposed approach achieves high accuracy rates of 99.84% and 96.4% for intra (subject by subject) and inter (cross-subject) modes, respectively. SVM emerges as the most effective model for drowsiness detection in the intra mode, while MLP demonstrates superior accuracy in the inter mode. This research offers a promising avenue for implementing proactive drowsiness detection systems to enhance occupational safety across various industries.

We are very thankful to the reviewers for their deep and interesting reviews. We have revised our research paper in light of their useful suggestions and comments.

Response to reviewer

Reviewer 2: The paper suffers from a significant lack of detailed information. Its discussion is too shallow.

I think the paper should be extensively modified.

Comment1

In the title, it’s written as ‘APproach’.

Response

Thank you for bringing this to our attention. We appreciate your keen observation. The title has been corrected to use standard capitalization, and "APproach" has been amended to "Approach".

Comment2

The authors filtered raw signals at 30 Hz. How can you say higher frequency components are irrelevant? While it is true that high frequency components have higher risks of containing artifacts, it doesn’t mean there are no relevant components in such high frequencies. The authors should justify this choice.

Response

An explanation justifying the choice to filter raw signals at 30 Hz, addressing concerns about higher frequency components, has been added in section 4 on pages 9 and 10. Relevant references supporting this justification have also been included as follows in the text below.:

« Filtering raw EEG signals at 30 Hz plays a crucial role in enhancing the quality of signal analysis by mitigating the influence of artifacts and noise prevalent in higher frequency ranges. While higher frequencies can contain relevant neural information, artifacts such as muscle activity or electrical interference [65] often obscure them. Previous research in EEG signal processing has consistently shown that the most pertinent neural activity for tasks like drowsiness detection typically resides within lower frequency bands, predominantly below 30 Hz [66]. Our decision to filter at this frequency aligns closely with the objective of capturing neural oscillations associated with drowsiness, which are known to manifest predominantly in these lower frequency ranges [67]. This approach is widely adopted in studies focusing on vigilance states, ensuring a balance between preserving essential signal information and minimizing the impact of noise and artifacts, thereby enhancing the robustness and reliability of our analytical outcomes [68]. »

 [65] LUCK, Steven J. An introduction to the event-related potential technique. MIT press, 2014.

[66] BUZSÁKI, György. Rhythms of the Brain. Oxford university press, 2006.

[67] DE GENNARO, Luigi, MARZANO, Cristina, FRATELLO, Fabiana, et al. The electroencephalographic fingerprint of sleep is genetically determined: a twin study. Annals of neurology, 2008, vol. 64, no 4, p. 455-460.

[68] KAPPENMAN, Emily S. et LUCK, Steven J. The effects of electrode impedance on data quality and statistical significance in ERP recordings. Psychophysiology, 2010, vol. 47, no 5, p. 888-904.

Comment3

The authors should define acronyms at their first appearance before using it (RPSD). In addition, the window size for welch’s method was not disclosed.

Response

The acronym "RPSD" has been defined at its first appearance in the manuscript. The full term "relative power spectral density" has been added at its first mention in Section 4 page 10.

The window size used for Welch's method will be disclosed in the methodology section to provide transparency and reproducibility. This information has been added to Section 4.1 page 11 as follows in the text below.:

 “In this work, the size of the welch window is fixed to 2 seconds. The choice of using 2-second windows for our analysis aims to strike a balance between computational efficiency and analytical depth. Longer windows typically offer better frequency resolution and statistical reliability, but they also require more computational resources and may lead to increased processing time. Conversely, shorter windows can reduce computational overhead but may sacrifice analytical depth due to the need to process numerous short segments of data.”

Comment4

For the 5.2,

The authors must disclose all parameters for each method. (For example, the authors did not even mention what kernel they used for SVM.)

Response

Thank you for your comment. We have addressed the concern by adding comprehensive details of the parameters and classification operations for each model in Section 5.2. These additions can be found on pages 15, 16, 17, and 18 of the manuscript. Specifically, the parameters and classification operations for SVM, KNN, NB, DT, and MLP models have been elaborated upon in detail.

A comprehensive description of the Support Vector Machine (SVM) model, with a specific focus on the Radial Basis Function (RBF) kernel and its associated hyperparameters, has been added to Section 5.2 on pages 15 and 16 in the text below :

“The main goal of the SVM is to find a hyperplane that separates the different classes with the widest possible margin. The SVM hyperplane equation is defined by ():

                                                                                        (15)                                                               

With  represents the weight vector, b is the bias and x is the feature vector.

For optimal separation, SVM maximizes the margin  by synchronizing  under the constraint represented by the following equation ():

                                                                                        (16)

                           with        (17)

Where (xi,yi) represents the training dataset, which is 1<i<N. x represents the characteristic vector extracted from the EEG signals, y indicates the corresponding vigilance status labels, and N is the number of data. Moreover, K is the kernel function of the SVM, Si is the vector support,  are the weights, and b is the bias. The SVM algorithm supports more than one kernel (linear, polynomial, RBF, sigmoid), and the choice of kernel always depends on the data type. Equation (18) represents SVM with the radial basis function (RBF) kernel:

                                                                            (18)

This kernel adeptly handles non-linear EEG data, capturing intricate brain activity patterns associated with drowsiness. The penalty factor C and the kernel factor  represent the SVM-RBF hyperparameters.”

The equation of the K-Nearest Neighbors (KNN) classification operation has been added in Section 5.2 on page 16. The modification is presented in the text below.

“ For the KNN classification operation, the new sample x is assigned to the majority class among its k nearest neighbors. The equation (20) represents the predicted class :

                                                                       (20)

                     Where  represents the set of the k nearest neighbors of x. “

A clear description of the Naive Bayes (NB) model has been incorporated into Section 5.2 on pages 16 and 17 in the text below.

“ The following equation (21) represents the Bayes theorem:

                                                                                   (21)

Where P(y|x) represents the probability of class y given features x, P(x|y) is the probability of features x given class y, P(y) is the probability of class y, and P(x) is the probability of features x.

For the classification operation, the NB algorithm calculates the probability of each class and chooses the one with the highest probability as presented by equation (22):

                                 (22)          “                                     

An explanation of the classification operation and parameters for the Decision Tree (DT) model in Section 5.2 on page 17. These additions provide readers with a clearer understanding of how the DT model works and the significance of its parameters in the classification process. The modification is presented in the text below.

“ For classification, the DT algorithm is based on impurity measurements, such as entropy or the Gini index. Entropy is given by equation (23) and the Gini index by equation (24):

                                                                          (23)

                                                                                  (24) “

we have supplemented Section 5.2 on pages 17 and 18 with an explanation of the classification operation and the parameters of the Multilayer Perceptron (MLP) model. Additionally, we have provided insights into the adaptability of MLP with EEG drowsiness detection, highlighting its efficacy in capturing complex patterns and non-linear relationships within EEG data. The modifications are presented in the text below.

“ The equation (25) presents the output layer:

                                                                               (25)

 is the weight of the connection between neuron  and neuron ,  is the input ,  is the bias of neuron j,  is the activation function.

The loss function for classification with c classes is often cross-entropy:

                                                                            (26)

Alternatively,  is the binary indicator if the example  belongs to the class .

 is the predicted probability that the example  belongs to the class .

MLPs can directly handle high-dimensional EEG data, learning complex relationships between features and drowsiness levels. They offer flexibility in modeling brain activity patterns, capable of capturing both linear and nonlinear relationships. However, careful hyperparameter tuning and regularization are essential to prevent overfitting and ensure robust performance in EEG drowsiness detection tasks.”

Comment5

There are two ‘5.1’.

Response

Thank you for bringing this to our attention. We apologize for the oversight. We have rectified the duplicate subsection numbers in Section 5.

We have reorganized Section 5 as follows:

5.1. Final Data

5.2. Classification Algorithms

5.3. Evaluation Metrics

Comment6

I think the authors also know that several ML algorithms’ performance depends on a lot of factors. So, just comparing them to conclude SVM is the best is barely able to generalize.

Response

The choice of the ML algorithms used in the study has been based on specific considerations, and this rationale has been added and detailed in section 5.2 pages 16 and 18 as follows :

“This kernel adeptly handles non-linear EEG data, capturing intricate brain activity patterns associated with drowsiness. The penalty factor C and the kernel factor  represent the SVM-RBF hyperparameters.”

“MLPs can directly handle high-dimensional EEG data, learning complex relationships between features and drowsiness levels. They offer flexibility in modeling brain activity patterns, capable of capturing both linear and nonlinear relationships. However, careful hyperparameter tuning and regularization are essential to prevent overfitting and ensure robust performance in EEG drowsiness detection tasks.”

Our goal in this study was to discover the most performant machine learning algorithms in two different classification modes: INTRA and INTER. The INTRA mode (subject-by-subject) allowed us to evaluate the performance of models within individual subjects, while the INTER mode (cross-subject and combined subject) aimed to assess the models' ability to generalize across different subjects.

In the INTRA mode (Section 6.2) page 21, we found that SVM-RBF exhibited high accuracy in detecting drowsiness compared to other models. This can be attributed to its capability to handle non-linear relationships present in EEG features. The modification is presented in the text below :

“The research shows that the SVM with the Radial Basis Function (RBF) kernel with penalty factor C=1 and kernel factor =0.4 is the most exact classifier for detecting drowsiness in the intra mode, with just two C3-C4 derivations and seven features picked by RFECV, with an overall accuracy of 99.85%.”

Conversely, in the INTER mode (Section 6.2) page 24, MLP emerged as the model with the highest accuracy in detecting drowsiness. The retropropagation between neurons in MLP facilitated the minimization of inter-subject effects, resulting in a more generalizable approach capable of detecting drowsiness across different subjects with consistent accuracy. The modification is presented in the text below :

“The maximum accuracy obtained after the use of the RFECV in the inter mode is that of MLP with nine input neurons, one hidden layer containing 100 neurons, and two output neurons. The ReLU activation function is used for the hidden layer, and Adam is used as the optimization algorithm with a batch size of 64 with a value of 96.4% with protocol P4 as shown in Table 10 with a number of characteristics selected by RFECV equal to nine features. “

Comment7

The authors also did not explain the use of features. What motivated the use of these features? For better generalization, a deliberate selection process is needed for features. Just putting everything and performing several algorithms without rationale for choosing the hyperparameters or something are likely to induce overfitting.

Response

The explanation of the choice of feature extraction tools has been added in section 4.1, page 10, as follows in the text below.:

“The choice of EEG features is indeed crucial for effective drowsiness detection, as each analysis domain offers unique insights into indicators of drowsiness. In this study, we focused on extracting EEG features from different domains time, frequency, and time-frequency using robust tools such as statistical features over time, the Welch method, and Discrete Wavelet Transform. These techniques were selected to enhance the sensitivity and the generalizability of our drowsiness detection approach [11].”

The choice of hyperparameters has been detailed in section 6.1 on page 21 as follows in the text below:

“SVM with the Radial Basis Function (RBF) kernel with penalty factor C=1 and kernel factor =0.4 is the most exact classifier for detecting drowsiness in the intra mode, with just two C3-C4 derivations and seven features picked by RFECV, with an overall accuracy of 99.85%. Grid search optimization is used to select optimal hyperparameters.”

The choice of hyperparameters has been detailed in section 6.2 on page 24 as follows in the text below:

“MLP with nine input neurons, one hidden layer containing 100 neurons, and two output neurons. The ReLU activation function is used for the hidden layer, and Adam is used as the optimization algorithm with a batch size of 64 with a value of 96.4% with protocol P4 as shown in Table 10 with a number of characteristics selected by RFECV equal to nine features. Grid search optimization is used to select optimal hyperparameters.”

Comment8

In the same vein, to me, it looks like the authors just used the channels simply because they were available in the dataset.

Response

Thank you for your observation. In this work, we utilized the DROZY database because it specifically focuses on drowsiness, providing a relevant and robust dataset for our study. While DROZY offers data from five EEG channels (Fz, Cz, C3, C4, Pz), our goal was to develop a drowsiness detection approach that is adaptable to real-world scenarios. Therefore, we opted to work with only two EEG channels instead of all five, aiming to simplify the setup and make it more practical for real-world applications.

To determine the most appropriate channels, we conducted a thorough comparison operation. This involved testing each channel individually and comparing their accuracy to select the most effective ones.

The details of this channel selection process and the results of our comparisons have been added in Section 6.1 on page 19. The modifications are presented in the text below :

“DROZY database offers data from five EEG channels (Fz, Cz, C3, C4, Pz Moreover, the use of five electrodes may not always be feasible in real-world conditions, especially in embedded systems where considerations such as energy consumption and size are critical. Many existing approaches use a single electrode to detect decreased alertness, but this approach can be unreliable if the electrode malfunctions or loses contact with the scalp. To overcome these challenges and ensure reliability, we focused on identifying the two most efficient EEG channels that minimize the number of electrodes while maintaining accuracy.”

This approach ensures that our method is both effective and feasible for real-world drowsiness detection applications, without relying on the availability of multiple EEG channels.

Comment9

What is the core finding of the study?

Response

The main objective of this study is to propose a generalized EEG-based drowsiness detection approach using a minimal number of EEG electrodes. The goal is to detect drowsiness across different subjects with consistent accuracy by minimizing inter-subject variability.

To achieve this, we followed a multi-step process:

  • We extracted relevant EEG features from different domains (time, frequency, and time-frequency) to maximize the number of drowsiness indicators. This comprehensive feature set helps in capturing the nuances of drowsiness across various subjects.
  • We employed an intelligent feature selection method called Recursive Feature Elimination with Cross-Validation (RFECV), which combines RFE and cross-validation based on the SVM-RBF algorithm. This method allowed us to retain only the most relevant EEG features related to drowsiness, thereby enhancing the generalizability of our approach.
  • We tested different machine learning algorithms in two classification modes (INTRA and INTER) to identify the model that best manages inter-subject effects and maintains high detection accuracy.

To avoid confusion, more details about the core finding of the study have been added in section 6.4 ("Discussion") on page 26 as follows in the text below :

« The main objective of this study is to propose a generalized EEG-based drowsiness detection approach using minimal EEG electrodes to ensure consistent accuracy across subjects. We extracted comprehensive EEG features (time, frequency, time-frequency) to capture drowsiness nuances. Intelligent feature selection via RFECV optimized relevant feature retention. Evaluation across INTRA and INTER modes identified robust machine learning models for managing inter-subject variability while maintaining high detection accuracy. »

Comment10

I cannot find a real discussion. The current 6.4 is just a repetition of results. I strongly suggest inserting an additional section for a real discussion.

Response

Thank you for your valuable remark. Section 6.4, "Discussion," has been modified based on your suggestions and can now be found on page 26. The modifications are presented in the text below :

“The main objective of this study is to propose a generalized EEG-based drowsiness detection approach using minimal EEG electrodes to ensure consistent accuracy across subjects. We extracted comprehensive EEG features (time, frequency, time-frequency) to capture drowsiness nuances. Intelligent feature selection via RFECV optimized relevant feature retention. Evaluation across INTRA and INTER modes identified robust machine learning models for managing inter-subject variability while maintaining high detection accuracy.

This study can serve as a foundational step in developing a drowsiness monitoring device base on EEG signals. Numerous studies have indicated the potential of detecting drowsiness using EEG signals.

In Table 11, we compare our proposed method with existing methods in INTRA mode. In [19], the authors utilized wavelet transform to extract TF features from 32 EEG channels, achieving an accuracy of 82% in drowsiness detection. The authors in [21] introduced an EEG-based approach for sleepiness detection, focusing solely on spectral characteristics extracted from 4 EEG channels, resulting in 78% accuracy. Another study in [23] employed spectral characteristics extracted from 32 electrodes and reported an accuracy of 92.6%. In [24], using the same "DROZY" database as ours, the authors extracted Hjorth parameters and employed MLP classifiers, achieving an accuracy of 90%.

In Table 12, we compare our approach with other recent studies focusing on drowsiness detection in INTER (cross-subject) mode. In [61], the authors utilize EEG entropy for drowsiness detection in INTER mode, employing a hybrid classifier (LR+ELM+LightGBM) to achieve 94% accuracy. In [62], researchers use 30 EEG channels to extract EEG spectral characteristics (relative power) and employ SVM as a classifier, achieving 68.64% accuracy in drowsiness detection. In [63], EEG spectral characteristics are deployed with a decision tree (DT) classifier, achieving an accuracy of 85.6% in INTER mode drowsiness detection. Finally, in [64], authors extract power spectral density (PSD) from 32 EEG channels and use SVM as a classifier, achieving an accuracy of 87.16% in drowsiness detection.

Table 11. Comparative analysis of the proposed method versus other systems in INTRA mode.

Ref

Feature extraction method

Classifier

Database

Electrodes

number

A(%)

P(%)

S(%)

F1(%)

[19]

WT

KNN

Private

32

82.08

78.84

87.71

83.27

[21]

FFT

SVM

Private

4

78.3

80.92

78.95

76.51

[23]

PSD

Neural network

EEG driver drowsiness dataset [57]

32

92.6

92.7

-

92.7

[24]

Hjorth

Parameters

MLP

DROZY

1

90

-

-

-

[26]

PSD

SVM

DROZY

5

96.4

[28]

TQWT

SVM

Sahloul University Hopital

1

94

-

94.08

-

Proposed methods

Statics / RPSD /

DWT

SVM

DROZY

2

99.85

99.87

99.8

99.5

Table 12. Comparative analysis of the proposed method versus other systems in INTER mode.

Ref

Feature extraction method

Classifier

Database

Electrodes

number

A(%)

P(%)

S(%)

F1(%)

[61]

Entropy

Hybrid classifier (LR+ELM+LightGBM)

Private

2

94.2

-

94

-

[62]

Relative Power

SVM

Multichannel_EEG_recordings_during_a_sustainedattention_driving_task_preprocessed_dataset

30

68.64

-

-

-

[63]

Spectral power

DT

Private

(HITEC University, Taxila, Pakistan)

1

85.6

89.7

-

87.6

[64]

PSD

SVM

Private

(North eastern UNIVERSITY)

32

87.16

-

-

-

[28]

TQWT

SVM

Sahloul University Hopital

1

89

-

89.37

-

Proposed methods

Statics / RPSD /

DWT

SVM

DROZY

2

96.4

96.9

95.87

96

We observe that our proposed approach demonstrates greater precision compared to the aforementioned works in both INTRA and INTER modes, achieving accuracies of 99.85% and 96.4%, respectively. This high performance is achieved using only two EEG electrodes and a minimal number of features.

This substantial difference in performance is mainly due to:

  • The Different EEG features extracted from various analysis domains such as time, frequency, and time-frequency (TF) help increase the number of indicators of drowsiness, thereby enhancing the accuracy and generalization of the approach.
  • The use of a 10-second sliding window helps maintain critical information about drowsiness, enabling more precise detection compared to a 30-second window. This choice significantly enhances the accuracy of drowsiness detection by capturing more immediate and relevant changes in the EEG signals. Consequently, the approach benefits from improved sensitivity to variations in drowsiness levels, resulting in a more reliable and effective monitoring system.
  • The intelligent feature selection layer, composed of RFECV based on SVM-RBF, instead of dimension reduction tools like PCA and KPCA, helps maintain only the most relevant features related to drowsiness. Additionally, the k-fold cross-validation technique helps to eliminate overfitting, ensuring the model's robustness and generalizability.
  • The selection of suitable EEG channels (C3, C4) with the highest precision helps minimize the effect of interference between electrodes, enhancing the system's accuracy and making it more adaptable to real-life conditions.”

Other recent works have been added to Table 12 to enrich the comparison in INTER mode. The references are presented in the text below :

[61]     MIN, Jianliang, XIONG, Chen, ZHANG, Yonggang, et al. Driver fatigue detection based on prefrontal EEG using mul-ti-entropy measures and hybrid model. Biomedical Signal Processing and Control, 2021, vol. 69, p. 102857.

[62]     CUI, Jian, LAN, Zirui, SOURINA, Olga, et al. EEG-based cross-subject driver drowsiness recognition with an interpretable convolutional neural network. IEEE Transactions on Neural Networks and Learning Systems, 2022.

[63]     ARIF, Saad, MUNAWAR, Saba, et ALI, Hashim. Driving drowsiness detection using spectral signatures of EEG-based neu-rophysiology. Frontiers in physiology, 2023, vol. 14, p. 1153268.

[64]     WANG, Fei, WU, Shichao, PING, Jingyu, et al. EEG driving fatigue detection with PDC-based brain functional network. IEEE Sensors Journal, 2021, vol. 21, no 9, p. 10811-10823.

Round 2

Reviewer 1 Report

Comments and Suggestions for Authors

All the concerns have been well revised, and thus this paper can be accepted for publication.

Author Response

Comment: All the concerns have been well revised, and thus this paper can be accepted for publication.

Response: Thank you for your thorough review and constructive feedback. We are pleased that the revisions meet your expectations. We appreciate your recommendation for publication.

Reviewer 2 Report

Comments and Suggestions for Authors

While the authors have made some improvements, some key areas remain unknown, particularly the level of detail in reporting machine learning parameters.

Specifically, the authors provided the kernel type (RBF), the penalty factor (C), and the kernel factor (γ). This is a good start, but more details such as the range of hyperparameters tested during optimization and the method of hyperparameter tuning (e.g., grid search, random search) would be necessary for full replicability. For k-NN, they mentioned the basic operation but did not specify crucial parameters like the value of k, the distance metric used, or any weighting scheme. Bayes also does not provide enough information, like details on the specific implementation (e.g., Gaussian, Multinomial) and any smoothing parameters. Decision tree also needs more details like the maximum depth of the tree, minimum samples per split, or any pruning strategies. MLP also needs details on important parameters like learning rate, batch size, number of epochs, dropout rates, and initialization methods.

I think discussion would benefit from potential variability and limitations of these models across different dataset. Also, discuss how the model parameters and performance might vary with different preprocessing techniques, feature extraction methods, and data quality. Include a section on robustness checks, such as sensitivity analysis of hyperparameters and feature selection criteria.

Author Response

Reviewer 2: While the authors have made some improvements, some key areas remain unknown, particularly the level of detail in reporting machine learning parameters.

Comment1

Specifically, the authors provided the kernel type (RBF), the penalty factor (C), and the kernel factor (γ). This is a good start, but more details such as the range of hyperparameters tested during optimization and the method of hyperparameter tuning (e.g., grid search, random search) would be necessary for full replicability.

Response

We appreciate the reviewer's suggestion for enhancing the replicability of our study. We have expanded the Methods section to include a detailed description of our hyperparameter optimization process for the SVM model. Details about hyperparameters selection for the SVM-RBF have been added in section 5.2, page 16, as indicated by the following text:

“For the selection of SVM-RBF hyperparameters in this work, a grid search strategy was employed, testing a range of values for the penalty factor C [0.1, 1, 10, 100] and the kernel factor γ [0.001, 0.01, 0.1, 1]. The optimal parameters were determined based on cross-validation, resulting in a penalty factor C of 1 and a kernel factor γ of 0.4, which provided the highest accuracy.”

Comment2

For k-NN, they mentioned the basic operation but did not specify crucial parameters like the value of k, the distance metric used, or any weighting scheme.

Response

Additional details about the value of k, the distance metric used, and the weighting scheme have been added in section 5.2 on pages 16 and 17, as indicated by the following text :

“Parameter k determines the number of nearest neighbors to consider when ranking an unknown data point. A low k-value makes the model more flexible but more sensitive to noise and a high k-value makes the model more robust to exceptions but can dilute local characteristics. For KNN distance metrics the most used metrics are:

  • Euclidean Distance

                                                                            (19)

  • Manhattan distance

                                                                             (20)

This distance measurement is used for feature spaces where the axes are more independent.

  • Minkowski Distance

                                                                         (21)

This measure represents a generalization of the Euclidean and Manhattan distances where p=2 corresponds to the Euclidean distance and p=1 to the Manhattan distance.

Several studies show that the Euclidean distance is the most used distance metrics [][].

In the KNN algorithm, weighting schemes are essential parameters that give more importance to closer neighbors, thus influencing the final classification or regression decision. Commonly used weighting schemes include:

  • Uniform weighting

For this type of weighting schemes, every neighbor is assigned the same weight regardless of its distance from the test point.

                                                                                            (22)

  • Inverse distance weighting

For this type of weighting schemes, Closer neighbors have more weight since they are presumed to be more representative of the target class or value.

                         =                                                                (23)

                          where  is a small value to avoid division by zero. This scheme reduces the influence of distant neighbors.

  • Kernel function weighting

Kernel functions allow weighting neighbors according to different function shapes. Commonly used kernels include Gaussian (24), triangular (25), and Epanechnikov (26).

                                                                                  (24)

                                                                (25)

                                                              (26)”

“Grid search was used in this work to select the KNN parameters. The selected distance metric was Euclidean, with the k values tested ranging from 1 to 20. The optimal number of neighbors selected was five, and the chosen weighting scheme was uniform weighting.”

Comment3

Bayes also does not provide enough information, like details on the specific implementation (e.g., Gaussian, Multinomial) and any smoothing parameters.

Response

Details about the specific implementation and smoothing parameters have been added in section 5.2 pages 18 and 19 as stated in the text below :

“The "naive" part of the Naive Bayes classifier comes from the assumption that the characteristics are independent given the class label. This means that the joint probability of the characteristics given to the class can be expressed as the product of the individual probabilities as mentioned in equation (29):

                                                                              (29)

X represents the features vector.

There are several types of Naive Bayes classifiers, depending on the distribution of features:

  • Gaussian NB

This type is used when the features are continuous and assumed to follow a Gaussian (normal) distribution. The likelihood of the feature is given by the equation (30):

                         exp (-)                                              (30)

Where  and are the mean and standard deviation of the characteristic  for class y.

  • Multinomial NB

This type is used for discrete data, often used for the classification of documents where characteristics represent frequencies or occurrences of words. The likelihood is given by equation (31):

                                                                            (31)

Where  is the total number of all characteristics for class y.

  • Bernoulli NB

This type of NB is used for binary/Boolean characteristics, where each feature represents the presence or absence of a characteristic. The likelihood is given by (32):

                                                              (32)

In practice, especially with the Naive Bayes Multinomial and Bernoulli, smoothing techniques such as Laplace smoothing (additive smoothing) are used to manage zero probabilities as mentioned in equation (33):

                                                                                  (33)

Where  is the number of features  in class y,  is the total number of all features in class y, n is the number of features, and  is the smoothing parameter.”

“The Naive Bayes model selected by the grid search in this work was Gaussian Naive Bayes (GNB). The smoothing parameter of the GNB was set to its default value of 1e-9.”

Comment 4

Decision tree also needs more details like the maximum depth of the tree, minimum samples per split, or any pruning strategies.

Response

We have revised the manuscript to include comprehensive details about the Decision Tree model. Details about the maximum depth of the tree, minimum samples per split, and pruning strategies have been added in section 5.2 pages 19 and 20, as stated in the text below :

“The most important parameters of DT are:

The maximum depth of the tree (). This parameter helps control the complexity of the tree and prevent overfitting.

                                                                                  (35)

                         Where Depth(T) presents the depth of tree T.

We find also the minimum number of samples (min_samples_split), this parameter is required to be at a leaf node. If a leaf has fewer samples than this value, it will not be split further.

                                                                       (36)

                         Where  is the number of samples at the current node.

In addition, we find the minimum number of samples (min_samples_leaf), this parameter is required for splitting an internal node. This ensures that a node is split only if it has sufficient samples.

                          min_samples_leaf                                         (37)

                         Where  is the number of samples at a leaf node.

DTs are also known by post-pruning which presents a technique that aims to reduce the size of the tree by eliminating parts that do not bring significant benefit to the prediction, This simplifies the model and improves its ability to generalize to new data. This method involves calculating a cost measure C(T) for each T subtree and selecting the one with the minimum cost. C(T) is presented by equation (38) :

                                                                   (38)

Impurity(T) measures the impurity of tree T, |leaves(T)∣ counts the number of leaves, and α is a regularization parameter that balances between tree complexity and predictive performance. “

“For the Decision Tree (DT) parameters selected by the grid search, the maximum depth was set to 5. The minimum number of samples required to split an internal node was fixed at 4, and the minimum number of samples required to be at a leaf node was fixed at 2

. The complexity parameter was set at 0.1.”

Comment 5

MLP also needs details on important parameters like learning rate, batch size, number of epochs, dropout rates, and initialization methods.

Response

We have updated the manuscript to include the specific details of MLP implementation. Details about learning rate, batch size, number of epochs, dropout rates, and initialization methods have been added in section 5.2 pages 20 and 21, as stated in the text below :

“Critical parameters within MLP encompass:

  • Learning Rate

The learning rate (η) controls the speed at which model weights are updated. The weight update is presented by equation (41):

                          η                                                     (41)

  • Batch Size

The batch size (m) determines the number of formation examples used to calculate the gradient loss function at each iteration.

  • Number of epochs

The number of epochs is the number of complete passages through the training dataset. Each epoch consists of updating the weights for each batch in the dataset.

  • Dropout Rate

Dropout is a regularization technique where a certain percentage of neurons are ignored during training to avoid over-fitting. The dropout mask is shown in equation (42):

                                                          (42)

  • Initialization Methods

Initializing weights is crucial for effective learning. The two most commonly used initialization methods for MLP are Xavier and He. They are presented respectively by equation (43) and (44):

                                                                          (43)

                                                                                      (44)

 represents the dimension of the input space for the weights to be initialized.  represents the dimension of the output space for the weights to be initialized.

 represents the distribution used in Xavier initialization, where weights are chosen randomly in a specified interval. N represents the distribution used in He initialization, where weights follow a zero-centered Gaussian distribution with an adapted variance.”

“For the MLP parameters selected by the grid search, the model consisted of nine input neurons, one hidden layer containing 100 neurons, and two output neurons. The ReLU activation function was used for the hidden layer, and Adam was used as the optimization algorithm with a batch size of 64.”

Comment 6

I think discussion would benefit from potential variability and limitations of these models across different dataset. Also, discuss how the model parameters and performance might vary with different preprocessing techniques, feature extraction methods, and data quality. Include a section on robustness checks, such as sensitivity analysis of hyperparameters and feature selection criteria.

Response

  • Thank you for your constructive comment. To discuss the variability and limitations of the proposed method across different datasets, we utilized the Sahloul University Hospital, Sousse Tunisie database, which represents our lab database. Currently, we are actively increasing the number of subjects in this database to enhance the variability of EEG data available for analysis. Details regarding this database and a comparison with the DROZY database detection precision have been included in section 6.4 on page 29, as shown in the text below:

“To evaluate the detection precision and generalization of the approach across various subjects, we compare the performance of the ML models of the proposed approach using different databases. The comparison database (Sahloul University Hospital) [28] consists of EEG signals collected from eight healthy subjects aged 21 to 25 with no history of alcoholism or drug use. These EEG signals were recorded at the Vigilance and Sleep Center of the Faculty of Medicine in Monastir, following an experimental protocol approved by our faculty's Ethics Committee. All participants signed an informed consent form, which included a brief description of the research involving human subjects, before starting the experiment. This database, containing 45 hours of EEG data related to drowsiness, is currently available upon request from the concerned author. The data collection protocol required subjects to wake up before 10:00 AM and spend approximately four hours completing the task. Each subject's record is represented by 19 EEG channels (Fp1, Fp2, F2, F3, Fz, F4, F8, T3, C3, Cz, C4, T4, T5, P3, Pz, P4, T6, O1, O2).

The drowsiness detection results of the proposed approach using Sahloul University Hospital data [28] in the INTRA mode with SVM-RBF show an accuracy of 96.87%, precision of 95.9%, sensitivity of 97.2%, and an F1-score of 96.5%. For the INTER mode, the MLP shows an accuracy of 92.5%, precision of 91.2%, sensitivity of 92.3%, and an F1-score of 93.1%. The inter-subject variability in the EEG data still impacts the results slightly. However, it is noteworthy that even when using diverse datasets, the approach maintains strong generalizability and precision in drowsiness detection. This resilience is attributed to the diversity of EEG features, robust feature selection methods, and appropriate classifier parameterization chosen for the analysis.”

In addition, we aim to test other databases related to drowsiness to enhance the precision and generalizability of our proposed approach. This perspective has been added to the conclusion as shown in the text below:

”Additionally, testing the approach on various drowsiness-related databases is a crucial phase in evaluating the generalizability, robustness, and accuracy of this drowsiness detection method, reflecting our ongoing efforts to explore its potential across diverse datasets.”

  • We have expanded our manuscript to discuss how model parameters and performance can vary with different preprocessing techniques, feature extraction methods, and data quality. Details have been added in section 6.4 pages 30 as stated in the text below :

“In [19], the authors segmented the filtered EEG into 1-second intervals before extracting time-frequency (TF) features from 32 EEG channels using wavelet transformation, achieving 82% accuracy in drowsiness detection using KNN as a classifier. For feature selection, they employed the Neighborhood Component Analysis (NCA). In [21], the authors introduced an EEG-based approach to detecting drowsiness, focusing only on spectral characteristics. They used Fast Fourier Transform (FFT) to extract frequency characteristics from 1-minute EEG segments from 4 EEG channels and employed SVM-RBF for classification, resulting in 78% accuracy. Principal Component Analysis (PCA) was used for feature selection in this work. Another study in [23] used spectral characteristics (PSDs) extracted by FFT from 3-second EEG segments from 32 electrodes, reporting 92.6% accuracy using a neural network (NN). In [24], using the same "DROZY" database as ours, the authors extracted Hjorth parameters from 2-second EEG segments from a single EEG channel (C3) and used MLP classifiers, achieving 90% accuracy. This work did not employ any feature selection method.”

“ In [61], the authors utilized EEG entropy extracted from 1-second EEG segments for drowsiness detection in INTER mode, employing a hybrid classifier (LR+ELM+LightGBM) to achieve 94% accuracy without using any feature selection method. In [62], researchers used 30 EEG channels to extract EEG spectral characteristics (relative power) from 3-second segments filtered with a band pass filter [0,50] Hz and employed SVM as a classifier, achieving 68.64% accuracy in drowsiness detection by using all the features without selection. In [63], EEG signals were filtered with a Butterworth low-pass filter and segmented into 1-second segments. EEG spectral characteristics were then extracted with FFT and deployed with a decision tree (DT) classifier, achieving an accuracy of 85.6% in INTER mode drowsiness detection. In this work, Minimum Redundancy Maximum Relevance (MRMR) was used to reduce feature dimensions. Finally, in [64], the authors extracted power spectral density (PSD) from 32 filtered EEG channels segmented into 1-second segments and used SVM as a classifier, achieving an accuracy of 87.16% in drowsiness detection without using any feature selection methods.”

Additionally, we have included robustness checks, such as sensitivity analysis of hyperparameters and feature selection criteria on pages 31 and 32 of our manuscript. as stated in the text below :

” with only 2 electrodes, minimizing interference between electrodes. Additionally, the size of the EEG segments directly influences decision-making. The choice of 10-second EEG segments for drowsiness detection is justified by an optimal balance between temporal resolution and stability of the extracted features. Unlike shorter 1-3 second segments, which can be too noisy and variable, 10-second segments capture relevant EEG trends without losing granularity. Compared to 1-minute segments, 10-second segments allow faster detection of drowsiness transitions while being long enough to ensure good representativeness of states. The quality of the EEG features also greatly influences the accuracy of the approach. Using different EEG features in various domains (time, frequency, TF) increases the system's capacity for drowsiness detection, whereas using a single type of feature limits system capacity.

The choice of classification model is a critical phase that directly affects the performance of the approach. In the INTRA mode, SVM-RBF demonstrates a significant ability to handle the nonlinearity, complexity, and high dimensionality of EEG features for each subject. On the other hand, MLP minimizes and overcomes the effect of inter-subject EEG variability, enhancing the generalization of the approach. This precision in machine learning models is attributed to the careful selection of hyperparameters for each classifier, underscoring the importance of the hyperparameter optimization phase using grid search. The feature selection method is crucial for enhancing the precision of drowsiness detection. The influence of the selection method on detection accuracy is evident in the cited works. Selection methods based on dimension reduction techniques such as PCA and MRMR tend to be less accurate in detecting drowsiness compared to RFECV, which employs a machine learning model to retain the most relevant features. For instance, NCA uses KNN as a selection model, which can be limited due to the nonlinearity of EEG data. In contrast, SVM-RBF used with RFECV better captures the complex, nonlinear patterns in EEG signals, leading to higher precision in our proposed approach for both INTRA and INTER modes.”
